# DRAGON: Distributional Rewards Optimize Diffusion Generative Models

**Yatong Bai**[†]
*University of California, Berkeley*

*yatong_bai@berkeley.edu*

**Jonah Casebeer**
*Adobe Research*

*jonah.casebeer@ieee.org*

**Somayeh Sojoudi**
*University of California, Berkeley*

*sojoudi@berkeley.edu*

**Nicholas J. Bryan**
*Adobe Research*

*njb@ieee.org*

**Reviewed on OpenReview:** *https://openreview.net/forum?id=gobhDku03J*

## Abstract

We present **D**istributional **R**ew**A**rds for **G**enerative **O**ptimizatio**N** (**DRAGON**), a versatile framework for fine-tuning media generation models towards a desired outcome. Compared with traditional reinforcement learning with human feedback (RLHF) or pairwise preference approaches such as direct preference optimization (DPO), DRAGON is more flexible. It can optimize reward functions that evaluate either individual examples or distributions of them, making it compatible with a broad spectrum of instance-wise, instance-to-distribution, and distribution-to-distribution rewards. Leveraging this versatility, we construct novel reward functions by selecting an encoder and a set of reference examples to create an exemplar distribution. When cross-modal encoders such as CLAP are used, the reference may be of a different modality (e.g., text versus audio). Then, DRAGON gathers online and on-policy generations, scores them with the reward function to construct a positive demonstration set and a negative set, and leverages the contrast between the two finite sets to approximate distributional reward optimization. For evaluation, we fine-tune an audio-domain text-to-music diffusion model with 20 reward functions, including a custom music aesthetics model, CLAP score, Vendi diversity, and Fréchet audio distance (FAD). We further compare instance-wise (per-song) and full-dataset FAD settings while ablating multiple FAD encoders and reference sets. Over all 20 target rewards, DRAGON achieves an 81.45% average win rate. Moreover, reward functions based on exemplar sets indeed enhance generations and are comparable to model-based rewards. With an appropriate exemplar set, DRAGON achieves a 60.95% human-voted music quality win rate without training on human preference annotations. As such, DRAGON exhibits a new approach to designing and optimizing reward functions for improving human-perceived quality. Example generations can be found at `https://ml-dragon.github.io/web`.

## 1 Introduction

Recent advances in diffusion models have transformed content generation across media domains, establishing new standards for generating high-quality images, video, and audio (Rombach et al., 2022; Ho et al., 2022; Liu et al., 2023; Ghosal et al., 2023). While these models achieve impressive results through sophisticated

[†]Work done as an intern at Adobe Research, supported in part by the U.S. Army Research Laboratory and the U.S. Army Research Office under Grant W911NF2010219, Office of Naval Research, and NSF.

training schemes, their optimization process typically focuses on metrics that may not align with downstream objectives or human preferences. This misalignment creates a fundamental challenge: how can we effectively steer these models toward desired output distributions or optimize them for specific performance metrics?

A prominent approach to address this challenge has been fine-tuning using instance-level feedback. Methods such as reinforcement learning from human feedback (RLHF) (Christiano et al., 2017; Williams, 1992; Schulman et al., 2017) and related methods like Direct Preference Optimization (DPO) (Rafailov et al., 2023) and Kahneman-Tversky Optimization (KTO) (Ethayarajh et al., 2024) leverage pre-trained reward models or large-scale, pairwise preference data collected offline to guide the optimization process. While effective, these approaches face several key challenges. Media like audio, music, and video are highly perceptual, making it hard and expensive to create reliable preference pairs (e.g. music (Cideron et al., 2024), in-the-wild audio (Liao et al., 2024)). Research in language models considered implementing criteria-based reward signals to alleviate the preference data challenges (DeepSeek-AI, 2025). However, similar approaches have been scarce for media creation, because its evaluation metrics often measure distributional properties like Fréchet embedding distance, diversity, and coverage, which existing reward optimization approaches like reinforcement learning could not handle. Furthermore, preference-based training methods like DPO (Rafailov et al., 2023) are constrained by the implicit reward functions hidden in their training data, making it difficult to adapt these approaches to new objectives or target distributions without collecting new preference data.

To address these limitations, we introduce **D**istributional **R**ew**A**rds for **G**enerative **O**ptimizatio**N** (**DRAGON**), a versatile framework for fine-tuning generative models towards a desired outcome or target distribution. DRAGON offers an alternative to existing reinforcement learning (RL) methods or pair-wise preference approaches for optimizing a broad spectrum of rewards, including instance-wise, instance-to-distribution, and distribution-to-distribution signals. As shown in Figure 1a, the key components of DRAGON are: 1) a pre-trained embedding extractor and a set of (possibly cross-modal) reference examples, 2) a reward-based scoring mechanism that creates positive and negative sets of on-policy online generations, and 3) an optimization process that leverages the contrast between the positive and negative finite sets to approximate distributional rewards. To our knowledge, DRAGON is the first practical algorithm that reliably optimizes our entire taxonomy of reward functions for generative models. We believe the ability to handle distribution-to-distribution rewards is particularly valuable since we can directly optimize generation quality metrics. Such reward functions are ubiquitous in generative model evaluation metrics (Fréchet embedding distance (Heusel et al., 2017), Kullback–Leibler (KL) divergence (Kullback & Leibler, 1951), Inception score (Salimans et al., 2016)). Moreover, leveraging DRAGON's unique versatility, we can construct new reward functions by simply collecting a set of ground-truth examples without human preference, drastically reducing the effort to construct reward signals.

We demonstrate DRAGON's effectiveness through comprehensive experiments with text-to-music diffusion transformers. Our evaluation incorporates multiple common music generation metrics: audio-text alignment (CLAP score) (Elizalde et al., 2023; Wu* et al., 2023), Fréchet audio distance (FAD) (Kilgour et al., 2019) evaluated across diverse reference sets and embedding models, Vendi score (Friedman & Dieng, 2023) for reference-free output diversity, and a custom-built human preference model for reference-free aesthetics scoring, totaling 20 reward functions. As shown in Figure 1b, DRAGON consistently improves over baselines across these diverse reward functions, achieving an average of 81.45% target reward win rate while generalizing the improvement across evaluation metrics. Via human listening tests, we show that DRAGON can achieve a 61.2% win rate in human-perceived music quality without training on human preference data.

The contributions of our work can be summarized as follows:

- We propose DRAGON, a versatile online, on-policy reward optimization framework for content generation models. DRAGON can optimize non-differentiable reward functions that evaluate either individual generations or a distribution of them.

- We propose a new approach to construct reward functions by simply selecting an embedding extractor and a set of examples to represent an exemplar distribution.

- We propose a human aesthetics preference model for AI-generated music and find that DRAGON can leverage it to improve human-perceived music quality with a small set of (1,676) human-rated clips.

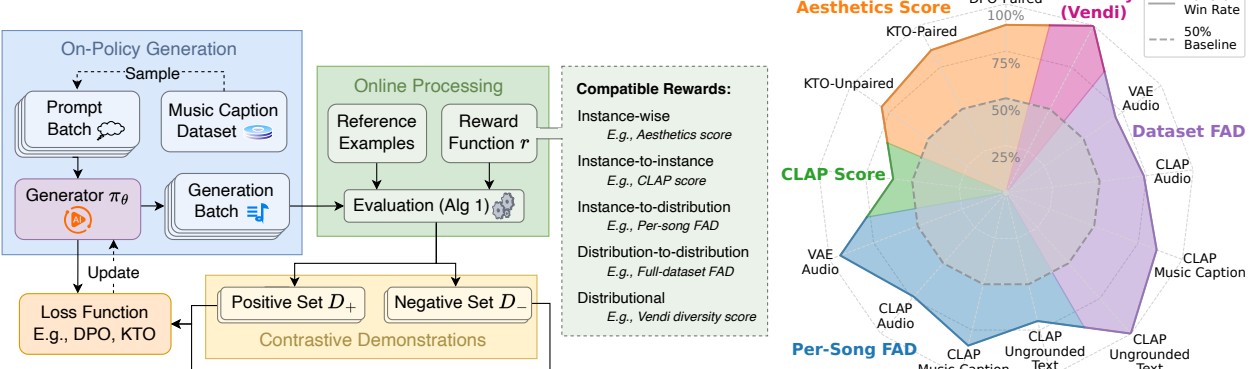

Figure 1a: An overall diagram of DRAGON, a versatile on-policy learning framework for media creation models. DRAGON is compatible with diffusion-based generation and can optimize various types of distributional reward functions.

Figure 1b: DRAGON significantly improves a full suite of rewards. Each vertex considers a reward metric and reports the win rate of the DRAGON model optimized for the metric.

- We analyze the relationship between FAD and human preference, and show that DRAGON can improve human-perceived music quality without human rating data by optimizing per-song or full-dataset FAD.
- We show that by leveraging cross-modal embedding spaces between text and music, DRAGON can improve music generation quality with text-only music descriptions without audio data.

While our experimentation focuses on music diffusion models, our framework is not specific to modality or modeling technique and can be applied to image or video generation as well as auto-regressive models.

## 2 Background and Related Work

### 2.1 Music Generation

Music generation has seen significant advances via auto-regressive token-based models (Vaswani et al., 2017; Zeghidour et al., 2021; Agostinelli et al., 2023; Borsos et al., 2023; Copet et al., 2023) and (latent) diffusion or flow-matching models (Novack et al., 2024a; Huang et al., 2023; Evans et al., 2024; Wu et al., 2024; Evans et al., 2025; Liu et al., 2024). Among auto-regressive approaches, MusicLM (Agostinelli et al., 2023) and MusicGen (Copet et al., 2023) are notable, with the former extended with RL via MusicRL (Cideron et al., 2024). Latent diffusion models like Stable Audio (Evans et al., 2024; 2025) show high-quality results and potential for ultra-fast generation (Bai et al., 2024; Novack et al., 2024b; 2025). However, reward optimization for diffusion models has proven more challenging than for auto-regressive ones (Wallace et al., 2024).

### 2.2 Diffusion Models

Diffusion models have emerged as a powerful paradigm able to generate high-quality and diverse samples (Sohl-Dickstein et al., 2015; Ho et al., 2020) via iterative de-noising. Such models commonly consist of a de-noising network $f_\theta$ that inputs noisy input $x_t$, diffusion time step $t$, and condition $c$ (e.g. text), and can either be discrete-time (Ho et al., 2020) or continuous-time score models (Song et al., 2021b; Karras et al., 2022). During training, given a demonstrative example $x_0$, we add Gaussian noise to form the noisy example $x_t$, and train $f_\theta$ to undo the noise injection. At inference time, generation begins with pure noise, from which $f_\theta$ is applied repeatedly to gradually denoise, forming a trajectory along the time step $t$ and noise level. Latent diffusion models (Rombach et al., 2022), where the above diffusion process takes place in a latent embedding space, have been leveraged across nearly all perceptual generation domains, including creating images (Rombach et al., 2022; Karras et al., 2022), video (Ho et al., 2022), speech (Popov et al., 2021; Eskimez et al., 2024), in-the-wild audio (Liu et al., 2023; Bai et al., 2024), and music (Huang et al., 2023; Forsgren & Martiros, 2022). Recent advances have shown that transformer-based architectures are advantageous, leading to diffusion transformers (DiT) (Peebles & Xie, 2023).

Table 1: Comparison of methods that optimize diffusion models to reward functions or human preferences.

| Category | Example Methods for Diffusion | Explicit Reward | Distributional Reward |
|---|---|---|---|
| RLHF | DDPO (Black et al., 2024), DPOK (Fan et al., 2024) | ✓ | ✗ |
| Preference Optimization | Diffusion-DPO (Wallace et al., 2024), Diffusion-KTO (Li et al., 2024) | ✗ | ✗ |
| DRAGON (ours) | DRAGON | ✓ | ✓ |

## 2.3 Reward Optimization for Diffusion Models

Diffusion models present unique challenges for reward-based optimization due to their iterative de-noising process. Existing work tackled this by formulating the diffusion process as a Markov Decision Process, with the reward signal assigned either exclusively to the final time step (Fan et al., 2024) or to all time steps (Black et al., 2024). More recent works explored the alternative of implicit reward optimization via learning from contrastive demonstrations. Diffusion-DPO (Wallace et al., 2024) and MaPO (Hong et al., 2024) optimize rewards defined with explicit binary preferences between paired samples (e.g., media contents with the same captions). Diffusion-KTO (Li et al., 2024) offers flexibility by being compatible with unpaired collections of preferred and non-preferred samples. Table 1 presents a comparison between DRAGON and these existing methods. Concurrent to our work, TangoFlux (Hung et al., 2024) represents an application of a CLAP score reward in diffusion modeling, proposing a semi-on-policy approach for in-the-wild audio generation.

## 2.4 Human Feedback Datasets and Aesthetics Models

Human feedback datasets and aesthetics models play a crucial role in guiding the optimization of generative models towards human preferences and reward models. Datasets such as SAC (Pressman et al., 2022), AVA (Murray et al., 2012), LAION-Aesthetics V2 (Schuhmann, 2022), Pick-a-Pic (Kirstain et al., 2023), and RichHF-18K (Liang et al., 2024) have been instrumental in improving the quality and alignment of generated images. These datasets are then used to train aesthetics models, which often map pre-extracted embeddings to a preference score. For audio/music generation, such an approach has been investigated with MusicRL (music) and BATON (in-the-wild audio), but is still rare and limited. In Section 4.1 and Appendix B, we describe the construction of our own aesthetics dataset and reward model for music generation.

## 3 Distributional Reward Optimization For Diffusion Models

We propose DRAGON, a reward optimization framework for optimizing diffusion models with a wide variety of reward signals as shown in Figure 1a. We consider a *distributional reward function* $r_{\text{dist}} : \mathcal{P} \to \mathbb{R}$ that assigns a reward value to a distribution of generations, where $\mathcal{P}$ is the set of all such distributions. We allow $r_{\text{dist}}$ to be non-differentiable (e.g., human preference). The outputs of our generative model $f_\theta$ form a distribution $\mathcal{D}_\theta$. When $f_\theta$ is a conditional generator, $\mathcal{D}_\theta$ depends on the distribution of conditioning $\mathcal{C}$, although we omit $\mathcal{C}$ for notation simplicity. Our goal is to fine-tune a pre-trained model via

$$\max_\theta \ r_{\text{dist}}(\mathcal{D}_\theta). \tag{1}$$

The formulation (1) includes traditional instance-level reward optimization widely studied in RLHF as a special case. Specifically, it is equivalent to instance-level optimization when the distributional reward $r_{\text{dist}}$ is the expectation of some instance-level reward $r_{\text{instance}}$, that is, $r_{\text{dist}}(\mathcal{D}_\theta) = \mathbb{E}_{X \sim \mathcal{D}_\theta}[r_{\text{instance}}(X)]$. However, unlike traditional RL, which is limited to instance-level rewards as it cannot distinguish between high and low quality generations without more granular feedback, DRAGON extends beyond this decomposable case.

To tackle such challenges and optimize $r_{\text{dist}}(\mathcal{D}_\theta)$, we construct a *positive demonstrative distribution* $\mathcal{D}_+$ such that $r_{\text{dist}}(\mathcal{D}_+) > r_{\text{dist}}(\mathcal{D}_\theta)$ and a *negative demonstrative distribution* $\mathcal{D}_-$. We then optimize model parameters $\theta$ to make $\mathcal{D}_\theta$ imitate $\mathcal{D}_+$ and repel $\mathcal{D}_-$. In the following sections, we address how to: 1) construct $\mathcal{D}_+$ and $\mathcal{D}_-$, and 2) optimize $\theta$ to make $\mathcal{D}_\theta$ imitate $\mathcal{D}_+$ and repel $\mathcal{D}_-$. Please also find pseudocode in Appendix F.

---

**Algorithm 1** Greedy algorithm for constructing $\mathcal{D}_+$ and $\mathcal{D}_-$ to optimize distributional reward $r_{\text{dist}}$.

1: Query $f_\theta$ twice to get finite sets of generations $\mathcal{D}_1 = \{x_{11}, \ldots, x_{1n}\}$ and $\mathcal{D}_2 = \{x_{21}, \ldots, x_{2n}\}$.
2: $(\mathcal{D}_+^{(0)}, \mathcal{D}_-^{(0)}) \leftarrow (\mathcal{D}_1, \mathcal{D}_2)$ if $r_{\text{dist}}(\mathcal{D}_1) > r_{\text{dist}}(\mathcal{D}_2)$ else $(\mathcal{D}_2, \mathcal{D}_1)$.
3: **for** $i = 0, 1, \ldots, n$ **do**
4:     At step $i$, swap the $i^{\text{th}}$ generation pair in $\mathcal{D}_+^{(i)}$ and $\mathcal{D}_-^{(i)}$ to form $\mathcal{D}_+^{'(i)}$ and $\mathcal{D}_-^{'(i)}$.
5:     $(\mathcal{D}_+^{(i+1)}, \mathcal{D}_-^{(i+1)}) \leftarrow (\mathcal{D}_+^{(i)}, \mathcal{D}_-^{(i)})$ if $r_{\text{dist}}(\mathcal{D}_+^{(i)}) > r_{\text{dist}}(\mathcal{D}_+^{'(i)})$ else $(\mathcal{D}_+^{'(i)}, \mathcal{D}_-^{'(i)})$.
6: **end for**
7: The final $(\mathcal{D}_+, \mathcal{D}_-)$ result is $(\mathcal{D}_+^{(n+1)}, \mathcal{D}_-^{(n+1)})$.

---

### 3.1 On-Policy Construction of $\mathcal{D}_+$ and $\mathcal{D}_-$

Existing work often assumes demonstrations $\mathcal{D}_+$ and $\mathcal{D}_-$ to be provided in advance (offline and off-policy), with $\mathcal{D}_+$ known to achieve a higher reward than $\mathcal{D}_-$ (Rafailov et al., 2023; Ethayarajh et al., 2024; Majumder et al., 2024). For instance-level rewards, this can be achieved by splitting a dataset into two halves at a reward threshold, a common RLHF approach. The limitations of off-policy learning necessitate a shift toward on-policy learning for several key reasons. First, for non-instance-level rewards such as FAD, the split becomes less straightforward. Second, large-scale offline data, required for effective off-policy learning, may be unavailable in practice. Third, on-policy optimization disentangles reward from dataset, providing more flexibility in reward choice. Finally, on-policy learning has demonstrated superior effectiveness and robustness because it enables real-time feedback and reduces data-policy mismatch. In the context of generative modeling, Tajwar et al. (2024) showed on-policy data helps language models learn from negative examples, and Hung et al. (2024) showed that even a partially on-policy data collection pipeline improves diffusion models. Hence, we focus on an online and on-policy approach, although offline data can be optionally incorporated into DRAGON with minimal algorithmic changes.

To construct on-policy distributions $\mathcal{D}_+$ and $\mathcal{D}_-$, we sample from $\mathcal{D}_\theta$ (which updates throughout training) online and query the distributional reward $r_{\text{dist}}$ on the fly. Specifically, before each training step, we collect two batches of observations from $\mathcal{D}_\theta$ by running full model inference with $f_\theta$ and denote them as $\mathcal{D}_1$ and $\mathcal{D}_2$. While the notations $\mathcal{D}_1, \mathcal{D}_2, \mathcal{D}_+,$ and $\mathcal{D}_-$ technically represent distributions of generations, practical computations of $r_{\text{dist}}$ use sampled sets as approximations. Hence, with a slight abuse of notation, we also use these notations to denote the sampled demonstration sets. For the special case where the distributional reward $r_{\text{dist}}$ can be decomposed into some instance-level reward $r_{\text{instance}}$, that is, $r_{\text{dist}}(\mathcal{D}_\theta) = \mathbb{E}_{X \sim \mathcal{D}_\theta}[r_{\text{instance}}(X)]$, the contrastive demonstration sets $(\mathcal{D}_+, \mathcal{D}_-)$ can be directly constructed by taking the better/worse halves of the union $\mathcal{D}_1 \cup \mathcal{D}_2$. This split can be determined by protocols such as element-level pair-wise comparison (if $\mathcal{D}_1$ and $\mathcal{D}_2$ consist of paired examples) or comparison with the batch median reward.

Optimizing general non-decomposable rewards like $r_{\text{dist}}$ that evaluate distributions of generations is more delicate as we need to disentangle each element's contribution. Exactly attributing each element's contribution is combinatorially expensive and impractical to do at training time for each batch. To this end, we propose Algorithm 1, a greedy algorithm. We initialize the positive/negative demonstration sets $(\mathcal{D}_+^{(0)}, \mathcal{D}_-^{(0)})$ with the higher/lower reward batch between $\mathcal{D}_1$ and $\mathcal{D}_2$. Next, we iteratively improve $\mathcal{D}_+$ through a swapping procedure. First, we tentatively swap a generation in $\mathcal{D}_+^{(0)}$ with one in $\mathcal{D}_-^{(0)}$ to form updated sets $\mathcal{D}_+^{'(0)}$ and $\mathcal{D}_-^{'(0)}$. Then, if $\mathcal{D}_+^{'(0)}$ improves the reward over $\mathcal{D}_+^{(0)}$, then accept the swap and set $(\mathcal{D}_+^{(1)}, \mathcal{D}_-^{(1)})$ to $(\mathcal{D}_+^{'(0)}, \mathcal{D}_-^{'(0)})$. Otherwise, reject the swap and set $(\mathcal{D}_+^{(1)}, \mathcal{D}_-^{(1)})$ to $(\mathcal{D}_+^{(0)}, \mathcal{D}_-^{(0)})$. Repeating these steps, we obtain $(\mathcal{D}_+^{(i)}, \mathcal{D}_-^{(i)})$ for $i = 0, 1, \ldots$. When stopping conditions are met, we take the latest $(\mathcal{D}_+^{(i)}, \mathcal{D}_-^{(i)})$ as the final $(\mathcal{D}_+, \mathcal{D}_-)$ pair.

For simplicity, we maintain equal sizes for $\mathcal{D}_1, \mathcal{D}_2, \mathcal{D}_+,$ and $\mathcal{D}_-$ and define the stopping condition as a complete pass over $\mathcal{D}_1$ and $\mathcal{D}_2$. Because $\mathcal{D}_1$ and $\mathcal{D}_2$ are random roll-outs during training, we do not further randomize and swap the $i^{\text{th}}$ pair at the $i^{\text{th}}$ iteration. Each step of Algorithm 1 is guaranteed to improve or maintain $\mathcal{D}_+$'s reward. When the loss function (to be discussed in Section 3.2) requires paired demonstrations,

we use the same conditioning but different random seeds to generate $\mathcal{D}_1$ and $\mathcal{D}_2$. Subsequent swaps are also performed with same-conditioning pairs, ensuring paired elements in $\mathcal{D}_+$ and $\mathcal{D}_-$ to provide direct contrast.

For multi-GPU training parallelization, we broadcast all generations to each GPU and then only swap the indices originally present on each GPU. As a result, the initial sets $(\mathcal{D}_+^{(0)}, \mathcal{D}_-^{(0)})$ are identical across GPUs, but subsequent $(\mathcal{D}_+^{(i)}, \mathcal{D}_-^{(i)})$ may differ from different swapping indices. Even still, the copy of $\mathcal{D}_+$ per GPU is guaranteed to be as good as both $\mathcal{D}_1$ and $\mathcal{D}_2$. The pseudocode in Appendix F includes this parallelization.

### 3.2 Learning From $\mathcal{D}_+$ And $\mathcal{D}_-$

We now optimize the generator parameters $\theta$ to make $\mathcal{D}_\theta$ attract $\mathcal{D}_+$ and repel $\mathcal{D}_-$, mathematically formulating this conceptual goal as minimizing the KL divergence $\mathrm{KL}(\mathcal{D}_+ \| \mathcal{D}_\theta)$ and maximizing $\mathrm{KL}(\mathcal{D}_- \| \mathcal{D}_\theta)$. Intuitively, this means encouraging $\mathcal{D}_\theta$ to cover as much of $\mathcal{D}_+$ and as little of $\mathcal{D}_-$ as possible. Note that

$$\arg\min_\theta \mathrm{KL}(\mathcal{D}_+ \| \mathcal{D}_\theta) \;=\; \arg\max_\theta \int \log \pi_\theta(x_0)\, d\mathcal{D}_+(x_0) \;=\; \arg\max_\theta \mathbb{E}_{x_0 \sim \mathcal{D}_+}\left[\log \pi_\theta(x_0)\right], \quad (2)$$

where $\pi_\theta(\cdot)$ denotes the likelihood for the model to generate a given example, and the integral is over the support of $\mathcal{D}_+$. Similarly, finding $\arg\max_\theta \mathrm{KL}(\mathcal{D}_- \| \mathcal{D}_\theta)$ is equivalent to finding $\arg\min_\theta \mathbb{E}_{x_0 \sim \mathcal{D}_+}\left[\log \pi_\theta(x_0)\right]$. Hence, our goal is equivalent to maximizing the log-likelihood for the model to generate examples in $\mathcal{D}_+$ and minimizing that of $\mathcal{D}_-$. Diffusion models pose a greater optimization challenge than auto-regressive ones because $\pi_\theta(\cdot)$ is implicit, making directly optimizing the KL-based conceptual objective impractical. To this end, DRAGON steers the likelihoods via surrogate loss functions, including but not limited to the Diffusion-DPO loss (Wallace et al., 2024) and the Diffusion-KTO loss (Li et al., 2024). Diffusion-DPO requires paired contrastive demonstrations, whereas Diffusion-KTO is more flexible and accepts unpaired ones. While Diffusion-DPO and Diffusion-KTO previously specialized in offline learning from large-scale preference datasets, they become components of an on-policy framework that optimizes arbitrary rewards when integrated into DRAGON, where $\mathcal{D}_+$ and $\mathcal{D}_-$ continuously evolve with $\pi_\theta$. We provide mathematical details about these two loss functions in Appendix A.1 and empirically compare them in Section 5.2, where we also ablate between paired and unpaired demonstrations. In Appendix A.2, we discuss potential extensions beyond binary $\mathcal{D}_+$ and $\mathcal{D}_-$, compatibilities with alternative loss functions like GRPO (Shao et al., 2024) and DPOK (Fan et al., 2024), and relationships to reward-weighted regression.

In total, DRAGON follows the illustration in Figure 1a and offers multiple advantages over existing reward optimization methods. First, DRAGON extends to distributional rewards like $r_{\mathrm{dist}}$ that are hard/unstable to differentiate (e.g. FAD with large network backbones) via learning from contrastive demonstrations. Second, DRAGON allows for cross-modal supervision by learning from distributions rather than exact point-wise matches. This flexibility allows us to use cross-modal exemplar embeddings to construct rewards, even when the reference and generation modalities have substantial structural differences. In our experiments, we show that DRAGON can leverage text embeddings to improve music generation using only textual descriptions.

## 4 Reward Functions

### 4.1 Instance-Wise Reward – Human Preference Dataset and Aesthetics Score Predictor

One of the most popular reward optimization tasks for content generation is aligning with human feedback, where human preference is the reward. To this end, human ratings of AI-generated examples provide relevant in-distribution guidance, and are thus more effective than ratings of human-created contents. Although open-source human preference datasets of AI image generations exist (Kirstain et al., 2023; Schuhmann, 2022; Schuhmann & contributors, 2022; Murray et al., 2012; Pressman et al., 2022), such resources are extremely rare for music. To demonstrate DRAGON's ability to align music generations to human preferences, we collect a human rating dataset of AI-generated music and build a custom music aesthetics predictor model. This predictor serves dual purposes: a reward model for DRAGON to optimize, and an evaluation metric for evaluating DRAGON models trained for other reward functions.

Our human preference dataset, which we call Dynamo Music Aesthetics (DMA), consists of 800 prompts, 1,676 music pieces with various durations (total 15.97 hours), and 2,301 ratings from 63 raters on a scale of 1–5. The 1–5 rating scale makes our human feedback more fine-grained than binary pairwise comparison datasets such as (Kirstain et al., 2023). DMA dataset and collection details are reported in Appendix B.1.

Our aesthetics predictor consists of a pre-trained CLAP audio encoder and a kernel regression prediction head, which we train on the DMA dataset. The textual prompt is not shown to the predictor. To determine model implementation details (e.g., music pre-processing, label normalization, CLAP embedding hop length) to optimize model performance on unseen data, we use a train/validation dataset split to perform an ablation study, which is presented in Appendix B.2. With model generalization verified, we remove the train/validation split and use all data to train the final predictor. A subjective test verified that generations with high predicted aesthetics scores indeed sound better than those with low scores, demonstrating more authentic instruments and better musicality. When the aesthetics score assigned by the predictor model is used as the reward function, DRAGON requires no additional music data.

## 4.2 Instance-to-Instance Reward – CLAP Score

We use CLAP score (Wu* et al., 2023), a popular music evaluation metric (Cideron et al., 2024; Bai et al., 2024; Hung et al., 2024), to demonstrate DRAGON's capability to optimize instance-to-instance rewards. CLAP score is defined as the cosine similarity (clipped to be non-negative) between the CLAP embedding of a single generated audio instance and a single reference embedding. Leveraging CLAP's unified cross-modal audio-text embedding space, we use the CLAP text embedding of the matching textual prompt as the reference for each audio generation. Intuitively, higher CLAP score means higher quality and semantic similarity. When maximizing CLAP score, DRAGON only requires a set of prompts and does not need any human-created music. When optimizing aesthetics score or CLAP score, both of which assign reward values to individual generations, $\mathcal{D}_+$ and $\mathcal{D}_-$ are constructed via pair-wise comparison. We find that DRAGON can improve overall generation quality by optimizing CLAP score.

## 4.3 Distribution-to-Distribution Reward – Full-Dataset FAD

We use the Fréchet audio distance (FAD) to demonstrate how we can accommodate reward signals that compare distributions or sets of generation outputs (audio) to corresponding target distributions (audio or text). FAD (lower is better) is one of the most commonly used metrics for evaluating music generation models (Kilgour et al., 2019). Intuitively, FAD represents the difference between a generated music distribution and a reference distribution (often human-created music) in an embedding space. Suppose that $\mu_\theta$ and $\mu_{\text{ref}} \in \mathbb{R}^d$ are respectively the means of the embeddings associated with generated and reference examples. Similarly, let $\Sigma_\theta, \Sigma_{\text{ref}} \in \mathbb{S}_+^d$ denote the covariance matrices of the two distributions. FAD is computed as

$$\text{FAD}\big(\mu_\theta, \Sigma_\theta, \mu_{\text{ref}}, \Sigma_{\text{ref}}\big) \coloneqq \big\|\mu_\theta - \mu_{\text{ref}}\big\|_2^2 + \text{Trace}\left(\Sigma_\theta + \Sigma_{\text{ref}} - 2\big(\Sigma_\theta^{\frac{1}{2}} \Sigma_{\text{ref}} \Sigma_\theta^{\frac{1}{2}}\big)^{\frac{1}{2}}\right). \tag{3}$$

To minimize dataset FAD to match a reference distribution, we start by approximating the true generation distribution $\mathcal{D}_\theta$ with all generations in a training batch across all GPUs. Since full-dataset FAD assigns a reward value to a set of generations and does not rate each individual, $\mathcal{D}_+$ and $\mathcal{D}_-$ must be determined via Algorithm 1. I.e., the positive demonstrative set $\mathcal{D}_+$ is constructed with Algorithm 1 to have minimal dataset FAD. When multi-modal embedding spaces are used, the reference distribution can be in any supported modality. For example, when CLAP is used as the encoder, the reference can be either audio or text. When an audio embedding distribution is used as reference, DRAGON only requires the distribution's mean and covariance. When the reference is a text embedding distribution, no audio data is needed for supervision.

## 4.4 Instance-to-Distribution Reward – Per-Song FAD

In addition to using FAD for distribution-to-distribution rewards, we also use FAD in an instance-to-distribution setting. While FAD typically compares two distributions, it can also compare a single generation instance to a distribution by "bootstrapping" the instance. For music, we can split a generated waveform into shorter chunks and encode each chunk, forming a "per-song embedding distribution" (Gui et al., 2024).

We can then use (3) to compute the FAD between this single generation and the reference statistics. The reference statistics need not be per song and are computed using the entire reference dataset, and can again be in non-audio modalities if supported by the embedding space. Since per-song FAD is assigned to each example, DRAGON constructs the demonstration sets $(\mathcal{D}_+, \mathcal{D}_-)$ via element-wise comparison (same as aesthetics and CLAP score optimization). In the literature, per-song FAD has been used to predict audio quality and identify dataset outliers (Gui et al., 2024). We show that with DRAGON, music generation models can improve generation quality by directly minimizing per-song FAD.

### 4.5 Reference-Free Distributional Reward – Embedding Diversity (Vendi Score)

The Vendi score (Friedman & Dieng, 2023) is a diversity metric, for which a larger value means more diverse. Intuitively, a Vendi score of $v$ means that the diversity of a set of embeddings is similar to that of $v$ completely dissimilar vectors. To compute the Vendi score of given $n$ embeddings with dimension $d$ represented as a matrix $X \in \mathbb{R}^{n \times d}$, we first assemble an $n \times n$ positive semi-definite kernel matrix $K$. We use a linear kernel $K = \hat{X}\hat{X}^\top$, where $\hat{X}$ is obtained by normalizing $X$ so that each embedding has an $\ell_2$ norm of 1. Next, we compute the eigenvalues of $K$, denoted as $\lambda_1, \ldots, \lambda_n$. The Vendi score is then the eigenvalues' exponentiated entropy:

$$\mathrm{Vendi}(X) \coloneqq \exp\left(-\sum_{i=1}^{n} \lambda_i \log \lambda_i\right). \tag{4}$$

During training, Vendi score is computed over generations in each training batch. We demonstrate directly improving Vendi with Algorithm 1, a result only possible because DRAGON operates on distributions.

## 5 Experiments

### 5.1 Models, Datasets, Training Settings, and Evaluation Metrics

**Baseline model and pre-training.** We use the base diffusion model from Presto (Novack et al., 2025) to generate 32-second single-channel (mono) 44.1kHz audio. It includes a latent-space score-prediction denoising module (Rombach et al., 2022; Karras et al., 2022; Song et al., 2021b) based on DiT-XL (Peebles & Xie, 2023) that takes in the noise level and a text embedding as conditioning signals, a convolutional variational autoencoder (VAE) that converts audio to and from the diffusion latent space (Kumar et al., 2023), and a FLAN-T5-based text encoder (Chung et al., 2024a). The baseline model is pre-trained with diffusion loss to convergence on a 3600-hour instrumental music dataset with musical-metadata-grounded synthetic captions, which we call the Adobe Licensed Instrumental Music dataset (ALIM). Our inference uses 40 diffusion steps with the second-order DPM sampler (Lu et al., 2022) with CFG++ ($w = 0.8$) enabled in selected time steps (Chung et al., 2024b). Please see Appendix E for details.

**Training and evaluation prompts.** We use ALIM training prompts (same setting as in pre-training) for DRAGON fine-tuning. Our evaluation uses a combination of the captions in an independent ALIM test split (800 pieces), the captions in a non-vocal Song Describer subset (Manco et al., 2023) (585 pieces, abbreviated as SDNV), and the real-world user prompts in the DMA dataset (800 pieces). Unless specified otherwise, all evaluation metrics are computed with generations from these 2,185 prompts (one generation per prompt).

**Evaluation initial noise.** Diffusion models iteratively de-noise from random initializations, and hence their generations are highly dependent on the initial noise. For a deterministic and fair comparison, we hash each test prompt into a random seed, and sample the initial noise from this seed. As a result, the initial noises are different across prompts but identical for all models.

**Evaluation metrics.** Our evaluation metrics include the predicted aesthetics score, CLAP score, per-song FAD, full-dataset FAD, and Vendi diversity score. Since these metrics vary in numerical range and directionality, we report the win rate over the baseline model to ensure comparability. For dataset FAD, we sample 40-example generation subsets with replacement (the subset indices are the same for all models) 1000 times, compute the dataset FAD for each subset, and report the win rate among these 1000 results.

**Correlation measures.** In addition to evaluating and comparing DRAGON models, we offer correlation analyses between various reward signals. We quantify correlation with overall Pearson correlation (PLCC) and per-prompt Spearman's rank correlation (SRCC) in Appendix B.2.

**FAD encoders.** The per-song and dataset FAD use ALIM and SDNV as reference statistics. DRAGON training uses ALIM's training split statistics to compute FAD, while evaluation uses the test split. We consider CLAP and the diffusion VAE as FAD encoders, with the former known to provide high-quality semantic embeddings (Gui et al., 2024). To account for any VAE reconstruction inaccuracies, we decode the generated VAE embeddings to audio and back. Compared with CLAP, the VAE encoder produces

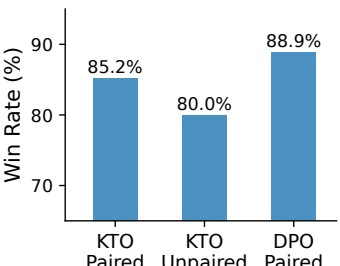

Figure 2: DPO versus KTO loss function; paired versus unpaired demonstrations.

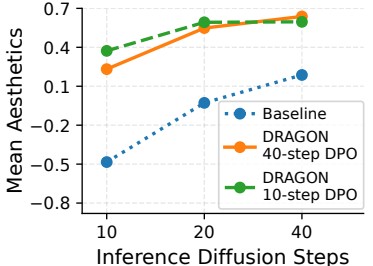

Figure 3: DRAGON with different demonstration diffusion steps and inference steps.

considerably more embeddings per song (1,836 versus 9) in a much lower dimension (32 versus 512).

**CLAP embedding setting.** We use the music-specializing LAION-CLAP checkpoint (Wu* et al., 2023; Chen et al., 2022). Since CLAP takes in 10-second 48kHz audio whereas our generations are 32-second 44.1kHz, pre-/post-processing is required. We focus on two settings: MA and FADTK. The MA setting is optimized for our music aesthetics predictor with ablation studies in Appendix B.2, while the FADTK setting follows (Gui et al., 2024). More details about and comparisons between the two settings are in Appendix C.2.

**Open-source comparison.** We compare DRAGON with open-source models such as Stable Audio (Evans et al., 2025) and MusicGen (Copet et al., 2023) in Appendix D.

## 5.2 Optimizing Instance-Level Rewards – Predicted Aesthetics Score and CLAP Score

We first use DRAGON to optimize instance-wise (reference-free) and instance-to-instance (reference-based) rewards using our aesthetics model and CLAP score, respectively. As shown in Figure 2, when optimizing the aesthetics score, DRAGON consistently achieves at least 80% reward win rate over the baseline, validating that our aesthetics model encodes learnable information, and DRAGON has a strong optimization capability.

When optimizing CLAP score, DRAGON achieves a 60.1% CLAP score win rate and a 68.7% aesthetics score win rate. This result shows CLAP score can act as a surrogate when human ratings are unavailable, but directly optimizing human ratings when available is more effective. Such an observation aligns with our statistical analysis on CLAP score. Over the DMA dataset, CLAP score and human-provided aesthetics have a 0.194 overall PLCC and a 0.135 per-prompt SRCC, indicating a positive but weak correlation.

**DPO versus KTO; paired versus unpaired.** Using the aesthetics score as the reward function, we perform ablation studies on different loss functions described in Section 3.2. Specifically, we focus on DPO loss with paired demonstrations, KTO loss with paired demonstrations, and KTO loss with unpaired demonstrations. As shown in Figure 2, DPO-Paired slightly outperforms KTO-Paired. DPO sees more stable training, whereas KTO improves the model faster. While KTO is more flexible by allowing for unpaired demonstrations, KTO-Paired outperforms KTO-Unpaired. Hence, direct pair-wise contrasting signals are advantageous, and we should prefer paired demonstrations when available.

**Ablation on diffusion steps.** Reducing the number of diffusion steps significantly accelerates training-time online generation at the cost of demonstration quality, but how does this affect the model fine-tuned with DRAGON? As shown in Figure 3, reducing the training-time diffusion steps from 40 to 10 (but still generating test examples with 40 steps) only induces a tiny change in generation quality. If we also decrease the number of test-time inference steps to 10, then the model trained with 10-step demonstrations can even outperform the one trained with 40 steps. In all inference settings, both DRAGON models outperform the baseline, confirming the generalization of model improvement across diffusion step settings. Moreover, the DRAGON models have overall flatter quality-versus-steps curves, with their 10-step generations outperforming the baseline model's 40-step generations. Hence, to achieve a similar generation quality, DRAGON can significantly reduce inference-time computation, manifesting some properties of distilled diffusion models such as consistency models (Song et al., 2023; Bai et al., 2024; Novack et al., 2025).

Table 2: DRAGON's win rates across reward functions. The "reward win rate" and "reward before/after" columns evaluate the reward function to optimize, which is different for each model. Aesthetics score, CLAP score, and FAD are reported for all models. FAD evaluation considers the diffusion VAE encoder and the CLAP audio encoder, using audio embeddings from the ALIM and SDNV datasets as the reference statistics.

(a) Win rates of DRAGON models that optimize instance-wise or instance-to-distribution reward functions.

| Reward Optimized | Reward Win Rate | Reward Before/After | Individual Metric Win Rates | | | | | |
| | | | | | Per-Song FAD | | | |
| | | | Aesthetics Score | CLAP Score | VAE Encoder | | CLAP Encoder | |
| | | | | | ALIM | SDNV | ALIM | SDNV |
|---|---|---|---|---|---|---|---|---|
| Aesthetics | 85.2% | .187/.638 | 85.2% | 52.2% | 54.9% | 55.3% | 58.9% | 55.1% |
| CLAP-Score | 60.1% | .300/.317 | 68.7% | 60.1% | 64.9% | 61.4% | 65.2% | 54.2% |
| Per-Song VAE-FAD ALIM-Audio | 93.9% | 30.8/16.3 | 78.3% | 50.9% | 93.9% | 94.0% | 83.9% | 81.0% |
| Per-Song VAE-FAD SDNV-Audio | 66.4% | 31.1/28.4 | 51.3% | 49.0% | 63.5% | 66.4% | 48.3% | 42.8% |
| Per-Song CLAP-FAD ALIM-Audio | 73.6% | .947/.867 | 61.5% | 51.6% | 49.5% | 49.6% | 73.6% | 70.7% |
| Per-Song CLAP-FAD SDNV-Audio | 56.3% | .990/.973 | 49.1% | 46.0% | 35.7% | 33.1% | 54.0% | 56.3% |
| Per-Song CLAP-FAD ALIM-Text | 83.5% | 1.58/1.48 | 78.3% | 65.4% | 64.0% | 65.7% | 70.4% | 65.9% |
| Per-Song CLAP-FAD SDNV-Text | 70.1% | 1.56/1.53 | 49.7% | 55.7% | 57.6% | 57.1% | 63.8% | 58.2% |
| Per-Song CLAP-FAD Human-Text | 83.7% | 1.60/1.54 | 52.9% | 60.7% | 38.0% | 41.0% | 64.3% | 63.3% |
| Per-Song CLAP-FAD Mixtral-Text | 70.1% | 1.53/1.49 | 65.5% | 52.2% | 52.8% | 52.2% | 60.3% | 60.5% |

(b) Win rates of DRAGON models that optimize reward functions like $r_{\text{dist}}$ that evaluate distributions.

| Reward Optimized | Reward Win Rate | Reward Before/After | Individual Metric Win Rates | | | | | |
| | | | | | Dataset FAD | | | |
| | | | Aesthetics Score | CLAP Score | VAE Encoder | | CLAP Encoder | |
| | | | | | ALIM | SDNV | ALIM | SDNV |
|---|---|---|---|---|---|---|---|---|
| Dataset VAE-FAD ALIM-Audio | 70.5% | 8.26/7.58 | 51.4% | 49.7% | 70.5% | 58.7% | 1.0% | 1.5% |
| Dataset VAE-FAD SDNV-Audio | 59.4% | 8.30/8.05 | 42.8% | 47.8% | 61.9% | 59.4% | 0.0% | 0.2% |
| Dataset CLAP-FAD ALIM-Audio | 73.6% | .214/.207 | 58.3% | 45.7% | 61.5% | 50.2% | 73.5% | 29.9% |
| Dataset CLAP-FAD SDNV-Audio | 83.2% | .260/.251 | 47.7% | 48.8% | 0.0% | 0.0% | 42.4% | 83.2% |
| Dataset CLAP-FAD ALIM-Text | 85.4% | .983/.967 | 68.8% | 59.5% | 88.2% | 84.7% | 2.1% | 0.9% |
| Dataset CLAP-FAD SDNV-Text | 81.6% | .799/.788 | 41.4% | 52.4% | 1.2% | 1.9% | 6.2% | 5.8% |
| Dataset CLAP-FAD Human-Text | 98.4% | .837/.813 | 57.4% | 55.8% | 26.5% | 38.9% | 26.2% | 14.1% |
| Dataset CLAP-FAD Mixtral-Text | 99.8% | .832/.786 | 64.6% | 61.1% | 8.3% | 14.1% | 1.3% | 0.0% |

All subsequent experiments use the Diffusion-KTO loss function with paired 40-step demonstrations. Appendix B.3 presents additional analyses on aesthetics optimization.

## 5.3 Optimizing Instance-to-Distribution Reward – Per-Song FAD

We fine-tune our music generator to minimize per-song FAD. In Appendix C.1, we perform ablation studies to demonstrate the statistical correlation between per-song FAD and human aesthetic perception, providing a theoretical foundation for improving human-perceived music quality via optimizing per-song FAD. Table 2a presents the model performance when optimizing per-song FAD with different reference statistics.

We first consider using audio embeddings of human-created music as the FAD reference statistics. As shown in Table 2a, for CLAP and diffusion VAE encoders alike, when the reference is ALIM ground-truth embeddings, minimizing per-song FAD enhances the target reward as well as the aesthetics score. The VAE embeddings are particularly powerful – optimizing the per-song VAE-FAD (ALIM) not only achieves a 93.9% win rate in this metric, but also generalizes to multiple other metrics. Over all models, the improvement in the per-song FAD to ALIM statistics is highly correlated with the SDNV-FAD improvement. However, using SDNV statistics as the DRAGON optimization target is less effective, likely due to SDNV's less consistent quality and smaller size, as well as the mismatch between SDNV FAD reference and ALIM training prompts. We thus highlight the importance of using high-quality reference music and avoiding dataset mismatches. Overall, these results show **DRAGON can enhance music generation without human feedback**.

Next, we leverage the cross-modality nature of the CLAP embedding space and use text embeddings as the FAD reference statistics for generated audio. Notably, by minimizing the per-song FAD to ALIM captions' CLAP embeddings, DRAGON achieves all-around improvements across all metrics in Table 2a. Surprisingly, the cross-metric generalization even outperforms optimizing FAD with audio reference. While it seems counter-intuitive that text can be more helpful than music, this result is explainable. As shown in Appendix C.1, compared to per-song FAD to audio reference, the audio-to-text FAD with ALIM captions is more strongly correlated with human preference. DRAGON can similarly optimize the per-song text-FAD to SDNV captions. Similar to the audio-reference case, when steering toward SDNV captions, while the reward win rate is high and the improvement generalizes to per-song audio-FAD, other metrics benefit less.

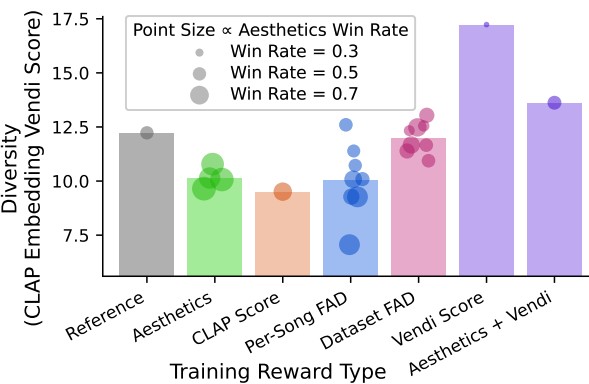

Figure 4: Vendi score of models optimized for each reward type. Point height represents Vendi score and point size represents aesthetics win rate. Each per-song/dataset FAD point train with a different reference statistic. Bar height averages point height.

Optimizing per-song text-FAD is similar to optimizing CLAP score in that both achieve aesthetic improvements with ALIM captions only, without any ground-truth music. In nearly all metrics, optimizing per-song text-FAD outperforms optimizing CLAP score. Hence, our novel instance-to-distribution pa-radigm is more effective than traditional instance-to-instance. In conclusion, **DRAGON can align music generation with human preference using high-quality captions without human-created music**.

Given DRAGON's capability of learning from text-only data, we experiment whether "ungrounded music descriptions" not associated with any music collection can also improve music generation. We specifically consider two sets of prompts: 800 human-created prompts in the DMA dataset, and 870 prompts generated by the Mixtral-8x7B large language model (LLM) (Jiang et al., 2024). To ensure LLM-generated prompt quality, we gather 10,506 initial prompts, and then use a greedy pruning algorithm similar to Algorithm 1 to minimize the dataset text-to-text CLAP-FAD with ALIM captions. As shown in Table 2a, when using human-created prompts, which are noisy and not based on actual music, as the exemplar set, DRAGON extracts learnable information via per-song FAD optimization, achieving an 83.7% reward win rate while improving CLAP score and per-song CLAP-audio-FAD. Similarly, the LLM-created prompts also encode learnable information and generalize even better to other metrics. Hence, we conclude that DRAGON can learn from ungrounded text-only music descriptions.

On average, DRAGON achieves an 81.4% win rate across all per-song FAD runs. Many runs (especially those using ALIM as the reference) improve the predicted aesthetics score, reaching an average aesthetics win rate of 59.9% without any human rating.

## 5.4 Optimizing Distribution-to-Distribution Reward – Full-Dataset FAD

Next, we leverage DRAGON's unique capability to learn from reward functions that evaluate distributions and minimize full-dataset FAD. We consider the same reference statistics as in the per-song FAD experiments in Section 5.3, and present the results in Table 2b. Optimizing dataset FAD is a particularly challenging task due to reward ambiguity and strong starting point performance. Specifically, learning from a reward like $r_{\text{dist}}$ that evaluates a distribution is harder and noisier than learning from an instance-level $r_{\text{instance}}$ due to the ambiguity of each instance's contribution to the reward. Meanwhile, because diffusion training implicitly matches the generated distribution to the true data, the baseline model already starts at a decent FAD.

Despite these challenges, all but one dataset FAD optimization run improve the reward function they optimize as shown in Table 2b. As with per-song FAD optimization, using the ALIM dataset as the reference statistics achieves multi-metric improvements, generalizing across FAD to different references and improving aesthetics score. This enhancement is observed for all three encoder-modality combinations (VAE, CLAP-

audio, CLAP-text), with the CLAP text embeddings attaining the best performance. Moreover, via dataset FAD optimization, ungrounded text embeddings (Human-Text and Mixtral-Text) can enhance music generation, especially in terms of aesthetics score and CLAP score. On average, dataset FAD optimization achieves a reward win rate of 81.5%, matching the average improvement for per-song FAD optimization. We thus conclude that **DRAGON can enhance diffusion models by directly minimizing dataset FAD**.

Compared to per-song FAD results, we observe several trends with dataset FAD optimization. First, VAE results are comparatively weaker and CLAP results are stronger. This makes sense because VAE embeddings are lower-dimensional, summarizing the entire generated distribution $\mathcal{D}_\theta$ with $\mu_\theta \in \mathbb{R}^{32}$ and $\Sigma_\theta \in \mathbb{S}_+^{32}$, totaling 1056 numbers. Hence, information is lost and optimization signals are weak. Second, cross-reference generalization is weaker. Improving per-song FAD to one reference statistic often means better per-song FAD to other references, a correlation not observed with dataset FAD. Recall that the references are always full-dataset even when the generated statistics are per-song. Hence, per-song FAD improvement may partially come from a per-song-versus-full-dataset gap shrinkage, which generalizes across reference statistics. This gap does not exist for dataset FAD, and hence imitating a distribution may imply moving away from others. Third, optimizing dataset FAD to SDNV often hurts other metrics. We believe this is because all training prompts are from ALIM, and hence using SDNV reference statistics induces confusion. Specifically, if we run DRAGON with ALIM prompts and ALIM reference statistics, then when tested on SDNV prompts, the FAD with respect to SDNV statistics improves (see Figure 10). However, if DRAGON pairs ALIM prompts with SDNV statistics, then we do not observe this improvement. In summary, with dataset FAD optimization, it is important to select a suitable encoder and a reference statistic that matches the prompt distribution.

Additionally, optimizing dataset FAD preserves more generation diversity than optimizing instance-level rewards like aesthetics, CLAP score, and per-song FAD. Figure 4 shows that optimizing per-song FAD worsens Vendi diversity score by an average of 17.7%, whereas optimizing dataset FAD only loses 3.2%. The next section will show that DRAGON can also explicitly optimize Vendi, balancing aesthetics and diversity.

### 5.5 Optimizing Reference-Free Distributional Reward – Vendi Diversity Score

To demonstrate DRAGON's capability to improve generation diversity, we select the CLAP embedding Vendi score as the reward function. During training, we compute the Vendi score across all demonstrations in each training batch. During evaluation, we compute Vendi score with all 2,185 test generations. As shown in Figure 4, when explicitly optimizing Vendi, DRAGON significantly increases the score, achieving a 40.84% relative improvement. However, since Vendi does not provide any music quality information, optimizing Vendi alone distorts the generations and hurts their aesthetics. To this end, we co-optimize Vendi score and aesthetics score by randomly selecting one of the two rewards at each training iteration with equal probability. The result is a model that simultaneously improves Vendi and aesthetics, producing diverse high-quality music. In summary, we find **DRAGON can promote generation diversity.**

### 5.6 Human Listening Test

We perform a listening test for subjective evaluation using two DRAGON models from Table 2 – one optimizing aesthetics score and the other optimizing per-song VAE-FAD to ALIM audio. We instruct 21 raters to compare the overall quality of blinded music pairs. Each rater is given 40 independently selected random SDNV prompts, along with the corresponding generation pairs. Out of each pair, one piece is from our baseline model and the other is from a DRAGON model (20 pairs for each DRAGON model). Clips are loudness-normalized to $-23$dB LUFS, and presented in random order. To accelerate the test, clips are randomly cropped into 5-second snippets (same start/end timestamp per pair).

Across all raters, both DRAGON models outperform the baseline, with DRAGON-aesthetics achieving a 60.2% human-labeled win rate and DRAGON-VAE-FAD managing 61.0%. Despite DRAGON-aesthetics receiving a higher machine-predicted aesthetics win rate (85.2%) than DRAGON-VAE-FAD (78.3%), its human-perceived quality is slightly worse. This is likely because DRAGON-aesthetics incurs some overfitting by directly maximizing the predicted aesthetics. In contrast, DRAGON-VAE-FAD learns in an instance-to-distribution approach, reducing overfitting by not relying on the DMA preference dataset. In summary, **DRAGON improves human-perceived music quality with sparse human feedback** (via our aes-

thetics model) **and even with no human feedback** (via per-song FAD). In Appendix B.4, we use statistical hypothesis testing to confirm our improvement and derive a 95% confidence win rate lower bound.

# 6 Conclusion

We presented DRAGON, a versatile reward optimization framework for content creation models that optimizes various generation quality metrics, along with a novel reward design paradigm based on exemplar sets. DRAGON gathers online and on-policy generations, uses the reward signal to construct a positive set and a negative set of demonstrations, and leverages their contrast to improve the model. In addition to traditional reward functions that assign a score to each individual generation, DRAGON can directly optimize metrics that assign a single value to a distribution of generations. Leveraging such flexibility, we constructed reward functions that match generations to a reference exemplar set in instance-to-instance, instance-to-distribution, and distribution-to-distribution formats. We evaluated DRAGON by fine-tuning text-to-music diffusion models with 20 reward functions, including a custom music aesthetics model trained on human preference, CLAP score, per-song FAD, full-dataset FAD, and Vendi diversity. Additionally, we provided ablation studies and analyzed the correlation between FAD and human preference. When optimizing the aesthetics score, DRAGON's win rate reaches up to 88.9%. When optimizing per-song/dataset FAD, DRAGON achieves an 81.4%/81.5% average win rate. By optimizing Vendi score, DRAGON improves generation diversity. Through listening tests, we show DRAGON improves human-perceived music quality at a 60.9% win rate with sparse or no human preference annotations, without additional high-quality music. In total, DRAGON exhibits a new approach to reward function optimization and offers a promising alternative for human-preference fine-tuning that lessens human data acquisition needs.

### Acknowledgments

We thank Ge Zhu, Zhepei Wang, Juan-Pablo Caceres, Ding Li, and anonymous raters who participated in constructing the DMA human preference dataset and evaluating the DRAGON models.

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

# Appendix

# A   Loss Functions for Learning from Demonstrations

## A.1   Diffusion-DPO and Diffusion-KTO Loss

Performing reward optimization for diffusion models can be more challenging than auto-regressive ones, because the likelihood $\pi_\theta$ is implicit and not directly available. To this end, we leverage the Gaussian assumption of diffusion models and the structure of the Gaussian density function (a squared, scaled, and exponentiated $\ell_2$ norm) to approximate the log-likelihood $\log \pi_\theta$ with an $\ell_2$ distance term.

Specifically, consider the forward diffusion process, where we inject Gaussian noise into a demonstration $x_0$ to form a noisy example $x_t$. That is, we randomly select a diffusion time step $t$ from a fixed distribution $\mathcal{T}$ supported on $[0, t_{\max}]$. Then, we sample $x_t$ from a Gaussian distribution $q(x_t|x_0) = \mathcal{N}(x_t; \alpha_t x_0, \sigma_t^2 I)$, where $\alpha_t \in [0, 1]$ and $\sigma_t$ represent the noise scheduling (Ho et al., 2020; Song et al., 2021a; Karras et al., 2022). Next, we query our model $f_\theta$ to denoise from $x_t$ and obtain the result $f_\theta(x_t, t)$, where we omit the conditional information (prompt) in the notation for simplicity. Wallace et al. (2024) showed $\log \pi_\theta(x_0)$ can be maximized via minimizing the surrogate objective function

$$\mathbb{E}_{t\sim\mathcal{T}, x_t\sim q(x_t|x_0)} \|x_0 - f_\theta(x_t, t)\|_2^2. \tag{5}$$

Similarly, $\log \pi_\theta(x_0)$ can be minimized by maximizing this surrogate objective.

Practical algorithms that optimize the above quantity often incorporate weighting and regularization terms to stabilize training (Yang et al., 2024; Wallace et al., 2024; Li et al., 2024; Hong et al., 2024). Among these algorithms, Diffusion-DPO and Diffusion-KTO are two of the most popular examples. Diffusion-DPO, which requires paired demonstrations, solves

$$\max_\theta \; \mathbb{E}_{\substack{(x_{+0}, x_{-0})\sim(\mathcal{D}_+, \mathcal{D}_-), \; t\sim\mathcal{T} \\ x_{+t}\sim q(x_{+t}|x_{+0}), \; x_{-t}\sim q(x_{-t}|x_{-0})}} \Big[ \sigma\Big( \beta(t) \cdot \big( A_\theta(x_{+0}, x_{+t}, t) - A_\theta(x_{-0}, x_{-t}, t) \big) \Big) \Big], \tag{6}$$

where

$$A_\theta(x_0, x_t, t) := \big\| x_0 - f_{\mathrm{ref}}(x_t, t) \big\|_2^2 - \big\| x_0 - f_\theta(x_t, t) \big\|_2^2. \tag{7}$$

Here, $A_\theta(x_0, x_t, t)$ represents how much $f_\theta$'s de-noising result is closer to the noiseless demonstration than $f_{\mathrm{ref}}$'s result, where $f_{\mathrm{ref}}$ is a "reference model" used for regularization. In practice, the pre-trained base model before DRAGON fine-tuning is used as $f_{\mathrm{ref}}$. Intuitively, (6) pulls the de-noising results from $f_\theta$ towards corresponding demonstrations in $\mathcal{D}_+$ and pushes them away from examples in $\mathcal{D}_-$.

When unpaired data is more accessible, Diffusion-KTO can be used. Diffusion-KTO is qualitatively similar to Diffusion-DPO, but allows for decoupling positive and negative demonstrations. Specifically, it optimizes the following objective:

$$\max_\theta \; \mathbb{E}_{\substack{\mathcal{D}_r\sim\mathrm{Bernoulli}(D_+, D_-) \\ t\sim\mathcal{T}, \; x_0\sim\mathcal{D}_r, \; x_t\sim q(x_t|x_0)}} \Big[ \sigma\Big( \beta(t) \cdot \mathrm{sgn}(\mathcal{D}_r = \mathcal{D}_+) \cdot \big( A_\theta(x_0, x_t, t) - \bar{A}_\theta \big) \Big) \Big], \tag{8}$$

where the distribution $\mathcal{D}_r$ is randomly chosen between $\mathcal{D}_+$ and $\mathcal{D}_-$, the binary variable $\mathrm{sgn}(\mathcal{D}_r = \mathcal{D}_+)$ is 1 when $\mathcal{D}_r$ is $\mathcal{D}_+$ and $-1$ when $\mathcal{D}_r$ is $\mathcal{D}_-$, the term $A_\theta(x_0, x_t, t)$ is defined in (7), and $\bar{A}_\theta$ is a regularization term. Specifically, $\bar{A}_\theta$ is obtained by sampling an independent batch of $x_0^{(1)}, \ldots, x_0^{(m)} \sim \mathcal{D}_r$, $t^{(1)}, \ldots, t^{(m)} \sim \mathcal{T}$, and $x_t^{(1)}, \ldots, x_t^{(m)} \sim q(x_t|x_0^{(1)}), \ldots, q(x_t|x_0^{(m)})$ and computing the "average $A_\theta(\cdot)$ value" via the formula

$$\bar{A}_\theta := \max\left( 0, \; \frac{1}{m} \sum_{i=1}^m A_\theta\big( x_0^{(i)}, x_t^{(i)}, t^{(i)} \big) \right).$$

In practice, due to the independence between examples in a training batch, each training batch itself can be used as a surrogate to compute $\bar{A}_\theta$ without additional explicit queries to the data loader.

In Section 5.2, we present ablation studies between Diffusion-KTO and Diffusion-DPO and between paired and unpaired demonstrations.

## A.2 Other Loss Functions and Beyond Binary $\mathcal{D}_+$ and $\mathcal{D}_-$

Using DRAGON, we can incorporate other loss functions that learn from binary demonstrations without modification. Beyond Diffusion-DPO (Wallace et al., 2024) and Diffusion-KTO (Li et al., 2024), such loss functions include MaPO (Hong et al., 2024) and D3PO (Yang et al., 2024).

DRAGON can also scale beyond binary demonstrations. For reward functions like $r_{\text{instance}}$ that evaluate individual generations, this extension is straightforward. Loss functions for binary demonstrations generally decide the direction of optimization via the positive/negative nature of a demonstration. That is, they explicitly or implicitly involve the $\text{sgn}(\mathcal{D}_r = \mathcal{D}_+)$ term introduced in the Diffusion-KTO loss (8). Since $\mathcal{D}_+$ and $\mathcal{D}_-$ are formed via a reward-based splitting operation, if we focus on a particular example $x_0$ sampled from the demonstrative distribution $\mathcal{D}_r$, then $\text{sgn}(\mathcal{D}_r = \mathcal{D}_+)$ is equivalent to $\text{sgn}(r_{\text{instance}}(x_0) > r_{\text{threshold}})$, where $r_{\text{threshold}}$ is the splitting threshold. We can extend beyond the binary preference assumption by replacing the discontinuous sign function with a continuous function, such as sigmoid or identity. One simple continuous-reward loss function is thus

$$\min_\theta \quad \mathbb{E}_{t \sim \mathcal{T}, \, x_0 \sim \mathcal{D}_\theta, \, x_t \sim q(x_t|x_0)} \left[ \beta(t) \cdot \left( r_{\text{instance}}(x_0) - r_{\text{threshold}} \right) \cdot \left\| x_0 - f_\theta(x_t, t) \right\|_2^2 \right], \tag{9}$$

which is equivalent to the DPOK objective with the KL-D setting (Fan et al., 2024), making it an example of reward-weighted regression algorithms (Peters & Schaal, 2007). That is, DPOK can be regarded as a special case of DRAGON under the instance-level reward scenario with non-binary preferences. One can similarly "continuize" the Diffusion-KTO loss as

$$\max_\theta \quad \mathbb{E}_{t \sim \mathcal{T}, \, x_0 \sim \mathcal{D}_\theta, \, x_t \sim q(x_t|x_0)} \left[ \sigma\left( \beta(t) \cdot \sigma\left( r_{\text{instance}}(x_0) - r_{\text{threshold}} \right) \cdot \left( A_\theta(x_0, x_t, t) - \bar{A}_\theta \right) \right) \right]. \tag{10}$$

Alternatively, one can consider other traditional RL loss functions for diffusion models like DDPO (Black et al., 2024) or derive variants of GRPO (Shao et al., 2024) specialized for diffusion models,

For reward functions like $r_{\text{dist}}$ that evaluate distributions, we can similarly replace $\text{sgn}(\mathcal{D}_r = \mathcal{D}_+)$ with some continuous transformation of $r_{\text{dist}}(\mathcal{D}_r)$. For example, the Diffusion-KTO loss can be "continuized" as

$$\max_\theta \quad \mathbb{E}_{\substack{\mathcal{D}_r \sim \text{Bernoulli}(D_+, D_-) \\ t \sim \mathcal{T}, \, x_0 \sim \mathcal{D}_r, \, x_t \sim q(x_t|x_0)}} \left[ \sigma\left( \beta(t) \cdot \sigma\left( r_{\text{dist}}(\mathcal{D}_r) - \tfrac{r_{\text{dist}}(\mathcal{D}_+) + r_{\text{dist}}(\mathcal{D}_-)}{2} \right) \cdot \left( A_\theta(x_0, x_t, t) - \bar{A}_\theta \right) \right) \right]. \tag{11}$$

It is also possible to scale the number of demonstrative distributions beyond two by modifying Algorithm 1.

# B Details and Ablations for Human Aesthetics Preference Alignment

## B.1 The DMA Music Preference Dataset

The collection pipeline of our DMA preference dataset is a two-phase process. In Phase 1, users interact with a collection of music generation models via an interface. After receiving the user prompt, the interface generates a piece, which the user rates on a scale of 1–5. In Phase 2, we reuse the user prompts from Phase 1 and generate additional music pieces. We provide four examples per prompt, which the user again rates on a scale of 1–5, providing a direct contrastive signal. To enhance data diversity, Phase 2 also randomly mixes in some LLM-created prompts and ALIM training set captions. During DRAGON fine-tuning, generation quality is expected to improve. To help our aesthetics model generalize and mitigate the likelihood for DRAGON to quickly become out of distribution, we additionally mix in some high-quality human-created music. Specifically, for ALIM prompts used in Phase 2, we randomly mix in ground-truth ALIM music.

Our final dataset consists of 800 prompts, 1,676 music pieces with various durations totaling 15.97 hours, and 2,301 ratings from 63 raters (multiple raters can rate the same generation). The proportion of each prompt source is shown in Table 3. Due to the small dataset size, we do not explicitly disentangle different aspects of music quality and text correspondence, and instead ask for a single overall opinion rating. Despite our dataset being orders of magnitude smaller than comparable modern image aesthetics datasets (see Table 4 for detailed comparisons), we will show that by leveraging DRAGON's versatile and on-policy training pipeline, we can improve human-perceived generation quality with high data efficiency.

Table 3: The DMA dataset's data sources, occurrences, and mean ratings of each source.

| Collection Phase | Prompt Source | Music Source | Occurrences | Mean Rating |
|---|---|---|---|---|
| Phase-1 | User prompts | Generated | 634 | 2.992 |
| Phase-2 | User prompts (reused) | Generated | 487 | 2.875 |
| Phase-2 | Training dataset captions | Generated | 361 | 3.277 |
| Phase-2 | LLM-generated prompts | Generated | 196 | 2.546 |
| Phase-2 | Training dataset captions | Human-created | 119 | 3.966 |
| Total | | | 1,676 | 2.919 |

Table 4: Comparison of aesthetic datasets across different modalities, sources, and rating scales.

| Dataset | Modality | Size | Content Source | Rating Source | Rating Scale |
|---|---|---|---|---|---|
| SAC (Pressman et al., 2022) | Image | >238,000 | AI-generated | Human-rated | 1-10 score |
| AVA (Murray et al., 2012) | Image | >250,000 | Human-created | Human-rated | 1-10 score |
| LAION-Aes V2 (Schuhmann, 2022) | Image | 1.2 Billion | Human-created | Model-predicted | 1-10 score |
| Pick-a-Pic (Kirstain et al., 2023) | Image | >1 Million | AI-generated | Human-rated | Paired binary |
| RichHF-18K (Liang et al., 2024) | Image | 18,000 | AI-generated | Human-rated | 1–5 multi-facet |
| BATON (Liao et al., 2024) | In-the-wild Audio | 2,763 | AI-generated | Human-rated | Paired binary |
| Audio-Alpaca (Majumder et al., 2024) | In-the-wild Audio | 15,000 | AI-generated | Model-predicted | Paired binary |
| MusicRL (Cideron et al., 2024) | Music | 285,000 | AI-generated | Human-rated | Paired binary |
| Ours | Music | 1,676 | Mostly AI-generated | Human-rated | 1–5 score |

## B.2 Aesthetics Predictor Details and Ablations

Our aesthetics predictor consists of a pre-trained CLAP audio encoder and a kernel regression layer as the prediction head. The textual prompt is not shown to the predictor. To determine model implementation details (e.g., music pre-processing, label normalization, CLAP embedding hop length) to optimize model performance on unseen data, we use a train/validation dataset split to perform an ablation study. With model generalization verified, we remove the train/validation split and use all data to train the final predictor. Finally, we perform a subjective test, verifying that generations with high predicted aesthetics scores indeed sound better than those with low scores, demonstrating more authentic instruments and better musicality.

To verify the performance of the aesthetics predictor and perform ablation studies, we split the DMA dataset into train/validation subsets by an 85/15 ratio. Specifically, we use overall PLCC and per-prompt SRCC to quantify the agreement between predicted aesthetics and human ratings on the validation split. PLCC is a number between $-1$ and $1$ that represents the "normalized covariance" (a larger value means more positive correlation and zero implies no correlation). SRCC is defined as the PLCC over rankings, and "per-prompt" means computing the SRCC among all generations from each prompt and averaging across all prompts. Per-prompt SRCC more accurately analyzes reward signals' agreement when comparing generations from the same prompt. Since we learn from contrastive demonstrations $\mathcal{D}_+$ and $\mathcal{D}_-$ evaluated with reward signals (often paired by prompt), per-prompt SRCC is particularly meaningful, as a higher per-prompt SRCC makes the aesthetics predictor more reliable as a reward function.

**Ablation 1: Embedding averaging.** As mentioned in Section 5.1, there are numerous ways to pre-process our 32-second 44.1kHz generations into the 10-second 48kHz format required by CLAP. We start with the original CLAP inference setting introduced in (Wu* et al., 2023) and explore several modifications:

1. **Average then rate** (original CLAP inference). For each music waveform, we take three random 10-second chunks an additional down-sampled chunk that captures global information, encode each chunk, average the four embeddings, and compute the aesthetics score with the average embedding. This setup is shared between training and inference for the aesthetics model.

2. **Rate then average.** During inference, we uniformly gather and encode four (partially overlapping) 10-second chunks, compute an aesthetics score from each embedding, and average the four scores. During training, we similarly extract up to four CLAP embeddings from each training music piece and train the aesthetics prediction head with the combined embedding dataset.

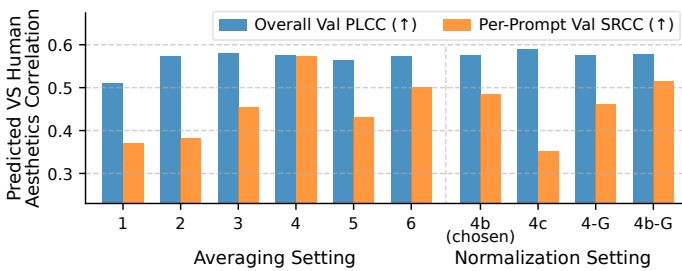 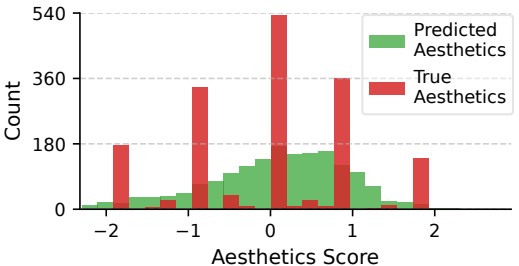

Figure 5: Ablation study on aesthetics model settings. Higher correlation with human ratings means better aesthetics model performance.

Figure 6: Histograms of human-rated and predicted aesthetics score over the DMA dataset after global label normalization.

3. **Setting 2 with down-sampled chunk.** On top of Setting 2, add the down-sampled global chunk from Setting 1 during inference (overall five embeddings). The model itself is the same as Setting 2.

4. **Setting 3 with 8+1 inference chunks.** Same as Setting 3, but increase the number of ordinary (non-down-sampled) chunks from four to eight (total nine chunks) during inference. The model itself is the same as Settings 2 and 3.

5. **Setting 3 with 16+1 inference chunks.** Same as Setting 3, but increase the number of ordinary inference chunks to 16 (total 17 embeddings). The model itself is the same as Settings 2, 3, and 4.

6. **Setting 4, but train with 8 embeddings.** Setting 4 uses four embeddings per music piece to train the aesthetics prediction head but use 8+1 embeddings during inference. Setting 6 increases the number of training embeddings to eight per piece to match the inference setting (thereby also increasing the size of the kernel matrix in the prediction head).

As shown in Figure 5, all settings obtain similar overall PLCC, with Setting 4 achieving the best per-prompt SRCC. Recall that the overall PLCC is computed across all music from all captions, entangling semantics and aesthetics, whereas per-prompt SRCC removes the influence of the semantic context by unifying the prompt. The low-SRCC settings tend to pay more attention to the semantics information dictated by the prompt and infer the aesthetics score via semantics-aesthetics correlation. In contrast, high-SRCC settings directly learn aesthetics information, making their outputs more closely aligned with human preference when ranking generations from the same prompt (which is the aesthetics model's task in DRAGON). In conclusion, while the regression quality is similar among all settings, different training/inference settings make the models learn different information. Hence, when training regressive aesthetics models, we need to look beyond typical full-dataset metrics and evaluate in settings that mirror a given use case. In our case, Setting 4 is the most reliable online music evaluation protocol.

Figure 5 also shows that the rate-then-average paradigm produces noticeably better results than CLAP's original average-then-rate approach proposed in (Wu* et al., 2023). This makes sense because when averaged across all pieces in the DMA dataset, the within-piece standard deviation among Setting 4's nine raw aesthetics predictions is 0.238. For reference, the dataset-wide aesthetics prediction standard deviation is 0.755, meaning that the within-piece variance is non-trivial. That is, different chunks of the same music piece can vary noticeably in predicted aesthetics score, and the rate-then-average approach mitigates this variance via ensembling, resulting in better performance.

Since Setting 4 achieves the best performance, we perform further ablation studies based on this setting.

**Ablation 2: Peak normalization.** While Setting 4 approximates human preference, it may have difficulty generalizing on all generated musical styles. That is, it may be possible for the music generator to "trick" the aesthetics model into assigning higher rewards by simple operations that do not improve human-perceived aesthetics. While it can be hard to completely generalize, we make efforts to block straightforward ones, specifically focusing on loudness sensitivity. We consider peak normalization, i.e., normalizing the waveforms to be within a specific range and focus on two normalization settings for the CLAP encoder:

4b. **Setting 4 with $[-0.5, 0.5]$ peak normalization.** Use the same training and inference setting as Setting 4, except we apply peak normalization to each 10-second chunk before encoding them, so that all waveforms are in $[-0.5, 0.5]$.

4c. **Setting 4 with $[-1, 1]$ peak normalization.** Same as Setting 4b, except the peak normalization scales each 10-second chunk to $[-1, 1]$ instead of $[-0.5, 0.5]$.

As shown in Figure 5, Setting 4b ($[-0.5, 0.5]$) achieves a much higher per-prompt SRCC than Setting 4c ($[-1, 1]$). While Setting 4b's per-prompt SRCC is slightly lower than Setting 4, we believe the magnitude-invariance guarantee outweighs the minor performance drop, and thus keep the peak normalization. Note that the diffusion VAE encoder, which doubles as a mapping to the latent diffusion space and an FAD reference embedding extractor, does not use peak normalization to preserve information for audio reconstruction.

**Ablation 3: Label normalization.** We perform label normalization to transform our 1–5 raw human ratings into a well-conditioned zero-mean distribution and study two settings:

1. **Settings 4 and 4b – Global label normalization.** The conventional normalization approach, which shifts the ratings by their population mean and scales them by their population standard deviation. The underlying assumption is that all labels are independent and form a standard Gaussian distribution.

2. **Settings 4-G and 4b-G – Per-rater normalization with Gibbs sampler.** Drawing inspiration from Hierarchical Bayesian Model (Lindley & Smith, 1972), we assume that the "rater harshness", quantified by a rater's average rating, forms a standard Gaussian distribution. The ratings from each rater then form another Gaussian distribution centered around this average. To effectively estimate the "average harshness" and the rating variance of each rater, we utilize the Gibbs sampler (Gelfand et al., 1990). Via rater harshness modeling, we can perform a more fine-grained per-rater normalization.

As shown in Figure 5, per-rater label normalization does not always outperform conventional global normalization, with 4b-G better than 4b but Setting 4 superior to Setting 4-G. While per-rater normalization is more sophisticated, a time-invariant Gaussian distribution may not accurately model each rater's ratings due to small sample size and potential preference change over time. As a result, global normalization, with the less precise overall Gaussian modeling, remains competitive.

Our final selection is Setting 4b, which reliably generalizes to unseen data while being simple. On the validation split of the DMA dataset, Setting 4b achieves an overall PLCC of 0.576 and a per-prompt SRCC of 0.484. For DRAGON training and evaluation, we remove the train/validation split and use Setting 4b to train a new model with all available data. Figure 6 presents this new model's prediction distribution over the DMA dataset.

**Comparison with other aesthetics models.** With Setting 4b, the performance of our preference model is on par with existing preference models of other modalities trained with much larger data sizes. For example, Liang et al. (2024) trained a sophisticated multi-branch image rating model that simultaneously predicts aesthetics score and several other metrics. With a similar 1–5 scoring scale, this model is one of the most comparable works to ours. Trained on about 16,000 rated images, the aesthetics prediction in (Liang et al., 2024) achieves a PLCC of 0.605. Despite our DMA preference dataset containing one order of magnitude fewer ratings, our model achieves a similar validation set PLCC of 0.576.

### B.3 Additional Aesthetics Score Optimization (RLHF) Analyses

As shown in Figure 7, when using DRAGON to optimize aesthetics score, the "middle-difficulty prompts", i.e., prompts on which the baseline model achieves a near-average aesthetics score, see the most significant improvement. This makes sense, because highly rated music pieces are comparatively rare in the DMA dataset. As a result, as the aesthetics score improves throughout DRAGON training, the aesthetics predictor gradually becomes less reliable, and further improvements become more challenging. To understand this, notice in Figure 7d that after DRAGON training, the model generations' aesthetics scores form a cluster whose most mass falls in the proximity of 0.9. As shown in Figure 6, the aesthetics model rarely makes predictions greater than 1 on its training dataset, and scores greater than 1.4 are almost never assigned. That is, DRAGON approaches the upper limit of the aesthetics model's useful output range, where the

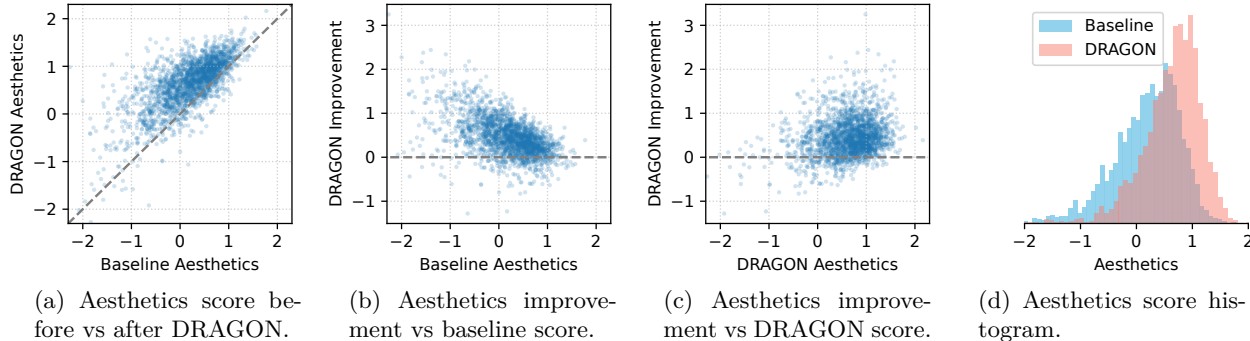

(a) Aesthetics score before vs after DRAGON.

(b) Aesthetics improvement vs baseline score.

(c) Aesthetics improvement vs DRAGON score.

(d) Aesthetics score histogram.

Figure 7: When optimizing aesthetics score, DRAGON improves low to medium-quality examples the most.

Table 5: Statistical analyses on human evaluation results.

| Model | Measured WR | 95% Conf WR Lower Bound | $p$-Value | $\mathbb{P}(\text{WR} > 50\%)$ |
|---|---|---|---|---|
| DRAGON-Aesthetics | 60.24% | 56.15% | $1.58 \times 10^{-5}$ | 99.9987% |
| DRAGON-VAE-PSFAD-ALIM | 60.95% | 56.87% | $4.15 \times 10^{-6}$ | 99.9997% |

aesthetics score becomes less reliable. Hence, if the aesthetics model can be improved in future work, we are confident that DRAGON optimization can further enhance music generation.

From a human preference alignment perspective, earlier image-domain approaches leveraged sparse human feedback (Pressman et al., 2022) or feedback on non-AI-generated examples that induce a distribution shift (Murray et al., 2012).[1] These weaker human preference signals are then augmented by training aesthetics predictors (Schuhmann & contributors, 2022) to create larger-scale synthetic preference datasets (Schuhmann, 2022). In some sense, DRAGON aesthetics score optimization follows a similar paradigm, but the creation of the synthetic dataset is now a part of the learning algorithm, and the demonstrations are on-policy. Hence, DRAGON sees less distribution shift and better utilizes sparse human preference. While more recent preference alignment methods leverage larger-scale human preference annotations for AI generation (Kirstain et al., 2023; Cideron et al., 2024), such data are highly expensive to collect, especially for the music modality where large-scale open-source preference datasets do not yet exist. We show that DRAGON addresses this challenge by learning from novel reward functions based on exemplar sets, achieving comparable results to directly learning from explicit human aesthetics ratings.

### B.4 Statistical Analyses on Human Listening Test Results

Following the collection of the DMA dataset, our listening test collects opinions about overall music quality, without disentangling different aspects of quality and prompt adherence. Based on the binary ratings from our listening test, we perform statistical analyses to obtain the following information:

- Whether we can reject the null hypothesis of "DRAGON does not improve upon the baseline".
- A 95% confidence lower bound for DRAGON's win rate.
- The posterior probability for DRAGON to outperform the baseline model (i.e. $> 50\%$ win rate).

Since we collect binary human preferences, we model the observed win rate as a binomial distribution $K \sim \text{Binomial}(n, w)$, where $n = 21 \times 20 = 420$ is the total number of preference annotations, $w$ is DRAGON's underlying true win rate which we aim to estimate, and the random variable $K$ is the number of positive ratings out of the $n$ data points. To evaluate the null hypothesis about DRAGON's performance, we perform a binomial test using our observed number of wins $k_\text{D}$. We present the resulting $p$-values in Table 5. Intuitively, our results mean that if the true win rate $w$ were to be no greater than 50%, then the probabilities of obtaining our measured win rate would be $1.58 \times 10^{-5}$ and $4.15 \times 10^{-6}$ for the two DRAGON models.

---

[1]While SAC now has over 238,000 images, it only had 4,000-5,000 when used to train the LAION aesthetics predictor.

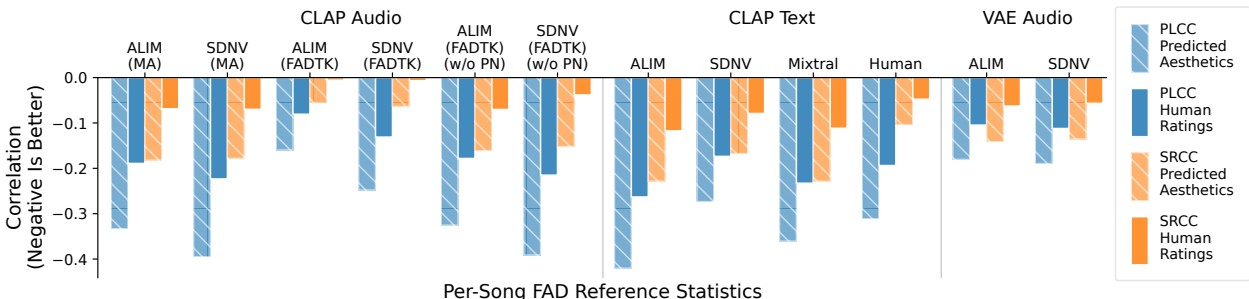

Figure 8: Correlation between per-song FAD with various reference statistics and aesthetics score. All numbers are negative because smaller is better for FAD whereas larger is better for aesthetics.

Since these are extremely small numbers, we can reject our null hypothesis with high confidence, concluding that DRAGON indeed outperforms the baseline.

To compute a one-sided 95% confidence lower bound, we solve for the largest $w$ value such that the probability of observing our positive count $\mathbb{P}_{K \sim \text{Binomial}(n,w)}(K \geq k_{\text{D}} \mid w)$, is no greater than 0.05. We consider the Clopper–Pearson confidence interval. Plugging in the binomial distribution mass function, we get

$$\mathbb{P}_{K \sim \text{Binomial}(n,w)}\big(K \geq k_{\text{D}} \mid w\big) = 1 - \sum_{i=0}^{k_{\text{D}}-1} \binom{n}{i} w^i (1-w)^{n-i} \leq 0.05.$$

The closed-form solution is $w_{\text{lower-bound}} = \text{Beta}^{-1}(\alpha, k_{\text{D}}, n - k_{\text{D}} + 1)$, where $\text{Beta}^{-1}$ is the inverse cumulative distribution function of the beta distribution. Plugging in our observed $k_{\text{D}}$ values for the two DRAGON in this test, we obtain the 95% confidence lower bound for $w$ shown in Table 5.

Finally, we use a Bayesian framework to analyze $\mathbb{P}(w > 0.5 \mid k_{\text{D}})$, the posterior probability for DRAGON's true win rate $w$ to be greater than 50%. Using a non-informative uniform prior $\text{Uniform}(0,1)$, we have the prior density function $p(w) \propto 1$. Via Bayes' Theorem, we have $p(w|k_{\text{D}}) \propto p(k_{\text{D}}|w) \, p(w)$. Substituting $p(k_{\text{D}}|w)$ with the binomial mass function, we get $p(w|k_{\text{D}}) \propto p^{k_{\text{D}}}(1-p)^{n-k_{\text{D}}}$, meaning that the posterior distribution is $\text{Beta}(k_{\text{D}}+1)(n-k_{\text{D}}+1)$. We can then use the cumulative distribution function of the beta distribution to compute $\mathbb{P}(w > 0.5 \mid k_{\text{D}})$. Table 5 shows that both DRAGON models have near-one posterior probability for $w > 0.5$, and thus we say with high confidence that DRAGON outperforms its baseline.

Note that the above analysis framework implicitly assumes that all binary preferences are independent. However, this is not strictly satisfied because the ratings are grouped by raters. To model the individual preference of each rater, we consider the alternative approach of treating each rater's observed win rate as a binomial variable. We then fit a generalized linear model (GLM) with a binomial family via logistic regression. Via the GLM approach, the 95% confidence lower bound for $w$ is 56.25% and 56.97% for the two DRAGON models, and the $p$-values are $1.55 \times 10^{-5}$ and $4.25 \times 10^{-6}$. These numbers are extremely close to the simple binomial modeling results shown in Table 5, meaning that while the independence assumption is not strictly satisfied, the effect of this inaccuracy is minuscule for our analyses.

## C  FAD Details and Ablations

### C.1  Per-Song FAD Correlation Analysis

Per-song FAD is a convenient statistical instance-to-distribution music quality metric. Since it is relatively new, its behavior has been less understood than more traditional metrics like human preference and dataset FAD. To motivate using per-song FAD as a reward function, we analyze its correlation with 1) human-labeled aesthetics score, 2) model-predicted aesthetics score, and 3) full-dataset FAD.

**Correlation between aesthetics scores and per-song FAD.** Similarly to how we evaluated the music aesthetics model, we use overall PLCC and the per-prompt SRCC to evaluate the agreement between per-song FAD and aesthetics score over the DMA dataset. We include both predicted aesthetics scores and raw human ratings in this analysis. Intuitively, overall PLCC measures the across-dataset correlation, whereas

per-prompt SRCC reflects "the degree to which ranking per-song FAD agrees with ranking aesthetics score," which is particularly meaningful for DRAGON training.

As shown in Figure 8, the per-song FAD that uses ALIM audio as the reference statistics is noticeably correlated with human preference. Hence, it is a valid music quality measure, and we can expect optimizing it to enhance the model. Note that per-song FAD's correlation with predicted aesthetics score is systematically more prominent than per-song FAD's correlation with human-provided aesthetics annotations, especially when per-song FAD uses the same CLAP embeddings in the aesthetics model. This observation suggests that the aesthetics model predictions are less noisy than human-predicted ones, but may

Table 6: Relationship between optimizing per-song and full-dataset FAD. The reference statistics come from the CLAP audio embeddings in the ALIM dataset.

| Generation Distribution | Per-Song FAD ($\downarrow$) | Dataset FAD ($\downarrow$) |
|---|---|---|
| $\mathcal{D}_{\text{ref}}$ | .947 | .106 |
| $\mathcal{D}_{\text{DRAGON-Aes}}$ | .920 | .116 |
| $\mathcal{D}_{\text{+persong}}$ | **.858** | .092 |
| $\mathcal{D}_{\text{+dataset}}$ | .903 | **.074** |

bring bias. Surprisingly, using ALIM's textual music descriptions' CLAP embeddings as the reference produces stronger correlations than using ALIM audio embeddings as the reference, implying that high-quality audio captions can offer rich and powerful insights into music quality. One explanation for text embeddings being more powerful than audio ones is that CLAP's text encoder, RoBERTa (Liu et al., 2019), was pre-trained with datasets that covered much wider topics than CLAP's audio encoder HTS-AT (Chen et al., 2022). Per-song text-FAD's high statistical correlation to human preference explains the high performance of the DRAGON model that optimizes this quantity (as shown in Table 2a), which enhances content creation without expensive human ratings based on text-only data.

**Connection between per-song and full-dataset FAD.** To understand the connection between per-song and full-dataset FAD, we analyze whether constructing $\mathcal{D}_+$ that optimizes one quantity also improves the other. We take the test generations from the baseline model and the DRAGON model that optimizes music aesthetics with the DPO loss, denoted as $\mathcal{D}_{\text{ref}}$ and $\mathcal{D}_{\text{DRAGON-Aes}}$ respectively, resulting in 2185 generation pairs. From each pair, we select the sample with a lower per-song FAD, and collect the 2185 chosen generations as $\mathcal{D}_{\text{+persong}}$. Meanwhile, we use Algorithm 1 to produce $\mathcal{D}_{\text{+dataset}}$, which also selects one piece from each of the 2185 pairs, but instead directly minimizes the full-dataset FAD.

Table 6 presents the per-song and full-dataset FAD of $\mathcal{D}_{\text{ref}}$, $\mathcal{D}_{\text{DRAGON-Aes}}$, $\mathcal{D}_{\text{+persong}}$, and $\mathcal{D}_{\text{+dataset}}$. We observe that the dataset FAD of $\mathcal{D}_{\text{+persong}}$ is lower than that of both $\mathcal{D}_{\text{ref}}$ and $\mathcal{D}_{\text{DRAGON-Aes}}$, but higher than $\mathcal{D}_{\text{+dataset}}$. Conversely, the average per-song FAD of $\mathcal{D}_{\text{+dataset}}$ is lower than both $\mathcal{D}_{\text{ref}}$ and $\mathcal{D}_{\text{DRAGON-Aes}}$, but higher than $\mathcal{D}_{\text{+persong}}$. We can thus conclude that per-song FAD is correlated with full-dataset FAD, but if our goal is to minimize one of these two metrics, then directly optimizing that metric is more powerful than using the other as a surrogate. This observation highlights the importance of DRAGON's unique capability to directly optimize reward functions that evaluate distributions, such as full-dataset FAD.

## C.2   MA Versus FADTK CLAP Embeddings for FAD Calculation

As discussed in Section 5.1, we consider two settings to compute CLAP embeddings for FAD calculation (for both per-song and full-dataset variants): MA and FADTK. The MA setting was determined via the ablation studies in Appendix B.2. It up-samples the 32-second 44.1kHz generations to 48kHz and uniformly splits them into eight partially overlapping 10-second audio chunks. The MA setting additionally down-samples the entire 32-second waveform to match the CLAP input sequence length and uses it to represent global information. Before encoding each chunk, we perform peak normalization to a range of $[-.5, .5]$, preventing reward hacking by merely manipulating the output magnitude. Eventually, the MA setting extracts nine 512-dimensional CLAP embeddings. The FADTK setting uses the same model checkpoint and up-sampling setting as MA. However, peak normalization is instead performed on the entire 32-second piece, which is subsequently divided into 10-second chunks with a 1-second hop length. As a result, FADTK produces more embeddings (25 instead of 9) per generation. The down-sampled chunk in the MA setting is not included.

For aesthetics score and CLAP score calculations, we obtain a score for each of the nine MA CLAP embeddings and average them. FAD (per-song and full-dataset) to text reference embeddings is also based on the MA embeddings due to their cross-modality nature shared with CLAP score. Meanwhile, since FAD

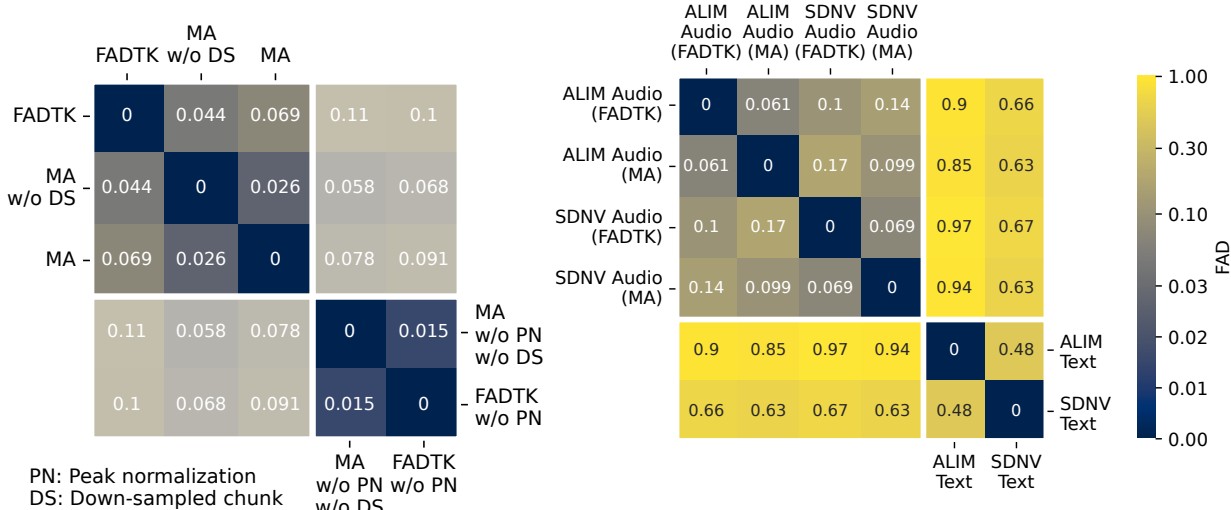

(a) Between FADTK and MA embeddings of SDNV music.

(b) Between different datasets and modalities.

Figure 9: FAD heatmap between different CLAP reference statistics.

to audio references is not cross-modal, we use the FADTK setting, which is reliable and standard in the literature. To compute Vendi score for a batch of music, we average the nine MA CLAP embeddings for each piece, and compute the Vendi score following (4) with the per-song mean embeddings in this batch.

To summarize, the differences between MA and FADTK are threefold:

- **Number of chunks/hop size.** The FADTK setting uses a one-second hop size (resulting in 25 chunks for our 32-second generations) whereas the MA setting uniformly samples 8 chunks.

- **Peak normalization.** The MA setting performs peak normalization for each chunk, whereas FADTK performs peak normalization before splitting into chunks.

- **Down-sampled global chunk.** Inspired by the original CLAP inference setting, the MA setting includes a down-sampled chunk to encode global information. The FADTK setting does not use this.

In this section, we ablate the influence of these three differences by encoding human-created music with various encoding settings and computing the FAD between the embedding distributions resulting from different settings. These FAD numbers are listed in Figure 9a. On the open-source SDNV dataset, we gradually modify the MA setting, removing the three main differences one by one to get closer to the FADTK setting, and examine how the FAD between the modified setting and the FADTK setting changes. The FAD between the unmodified MA setting and FADTK is almost as large as the FAD between two different audio datasets (SDNV-versus-ALIM) at 0.0688. When we remove the down-sampled block from the MA setting, the FAD decreases to 0.0439 but is still quite large. If we further remove peak normalization from both settings (now the only difference is hop size), then the FAD becomes only 0.0146. Comparing these numbers, we conclude that all three differences between FADTK and MA contribute to their discrepancies, with peak normalization asserting the strongest influence and hop size mattering the least.

In Figure 9b, we compare the SDNV and ALIM datasets' CLAP embedding distributions, considering FADTK and MA music embeddings as well as caption text embeddings. In both FADTK and MA settings, the FAD between SDNV and ALIM audio embeddings is about 0.1. The text embedding FAD between the two datasets is much larger at 0.48. In terms of audio-text correspondence, SDNV music descriptions are statistically closer to music pieces in the CLAP embedding space, with an FAD of around 0.65. In comparison, ALIM captions see a large FAD of nearly 1 from the corresponding audio data. For both ALIM and SDNV, MA audio embeddings are closer to the text embeddings. This observation supports using the MA setting when computing the FAD between generated audio and ground-truth text.

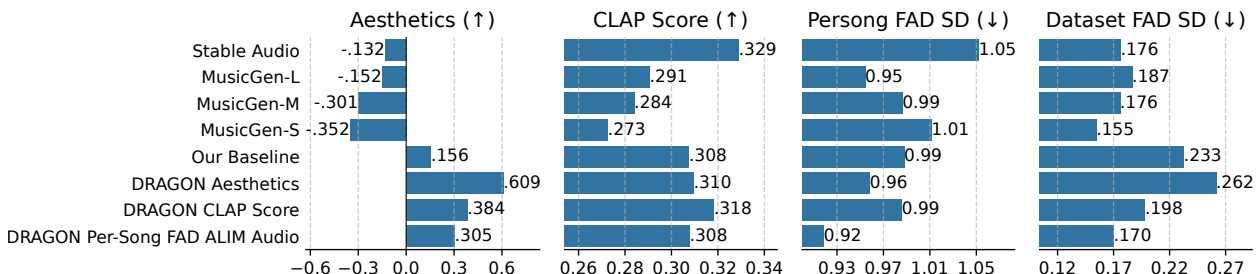

Figure 10: Comparing DRAGON and its pre-trained baseline model with open-source music generators.

Next, we analyze how the differences between FADTK and MA affect their role as per-song FAD reference statistics for evaluating generated music. As shown in Figure 8, when ALIM or SDNV audio embeddings are used as reference (the setting to encode generated music matches with the reference setting for consistency), the per-song FAD computed with the MA setting is more strongly correlated with human preference. In contrast, the FADTK setting sees a much lower PLCC and a near-zero per-prompt SRCC with human ratings, suggesting that per-song FADTK makes decisions almost entirely based on semantic information, rather than aesthetics. If we remove peak normalization from FADTK, then we recover some correlation with human feedback, implying that per-song FADTK also attends to loudness. Gui et al. (2024) proposed to use per-song FADTK to predict music quality and identify outliers, and our result shows that such capabilities may be a result of the correlation between aesthetics, loudness, and semantics. This makes sense, because Gui et al. (2024) showed that the quality ratings given by per-song FADTK (based on audio) and large language models (GPT-4, based on captions) strongly agree with each other.

In summary, even with the same encoder, the embedding statistics can significantly vary depending on the specific encode settings. When computing FAD, it is important to select a suitable setting and be consistent.

## D  Comparison with Open-Source Models

This section compares DRAGON with open-source models, specifically considering the following:

- MusicGen (Copet et al., 2023), a non-diffusion auto-regressive music generation model that predicts discrete audio tokens (Défossez et al., 2024) in sequence, coming in small, medium, and large variants.
- Stable Audio Open (Evans et al., 2025), a state-of-the-art audio diffusion model that generates variable durations of up to 45 seconds, following a similar design to our base model.

We use the non-vocal subset of Song Describer (SDNV) as the evaluation benchmark, and compare in aesthetics score, CLAP score (with text), per-song FAD, and dataset FAD metrics. Based on the results shown in Figure 10, we make the following observations:

- DRAGON and its pre-training baseline achieve higher aesthetics scores than Stable Audio and AudioGen. Since open-source models are out-of-distribution for our aesthetics model, there may be a bias. That said, among open-source models, the aesthetics model generally assigns higher scores to models known to be more capable.
- In terms of CLAP score, our baseline model is between Stable Audio and AudioGen. When we use DRAGON to optimize CLAP score, while the training prompts are from ALIM, the improvement generalizes to SDNV.
- The per-song FAD of our baseline model is in the best cohort. By explicitly optimizing per-song FAD via DRAGON, we achieve the state-of-the-art among the tested models. Again, while the training prompts and the FAD reference audio are all from ALIM, the improvement generalizes to SDNV.
- Using DRAGON to optimize CLAP score or per-song FAD also improves full-dataset FAD, with the DRAGON model that optimizes per-song FAD achieving one of the best dataset FAD numbers. Surprisingly, smaller MusicGen models achieve lower dataset FAD than larger ones in the family.

In summary, our baseline model is at least on par with state-of-the-art open-source models, and DRAGON can steer music generations to improve various performance metrics.

# E    Model Details and Hyperparameters

**Convolutional VAE.** We build on the Descript Audio Codec (DAC, or Improved RVQGAN) (Kumar et al., 2023) architecture and training scheme by using a KL-bottleneck with a dimension of 32 and an effective hop of 768 samples, resulting in an approximately 57Hz VAE. We train to convergence using the recommended mel-reconstruction loss and the least-squares GAN formulation with $\ell_1$ feature matching on multi-period and multi-band discriminators.

**Diffusion Transformer (DiT).** Following the base model in (Novack et al., 2025), our model backbone builds upon DiT-XL (Peebles & Xie, 2023), with modifications aimed at optimizing computational efficiency. Specifically, it uses a streamlined transformer block design, consisting of a single attention layer followed by a single feed-forward layer, similar to Llama (Llama Team, 2024). The diffusion hyperparameter design follows EDM (Karras et al., 2022), with $\sigma_{\mathrm{data}} = 0.5$, $P_{\mathrm{mean}} = -0.4$, $P_{\mathrm{std}} = 1.0$, $\sigma_{\mathrm{max}} = 80$, and $\sigma_{\mathrm{min}} = 0.002$. Also following EDM, we apply a logarithmic transformation to the noise levels, followed by sinusoidal embeddings. These processed noise-level embeddings are then combined and integrated into the DiT block through an adaptive layer normalization block. For text conditioning, we concatenate the T5-embedded text tokens with audio tokens at each attention layer. As a result, the audio token query attends to a concatenated sequence of audio and text keys, enabling the model to jointly extract relevant information from both modalities. Pre-training of the baseline diffusion model lasted five days across 32 Nvidia A100 GPUs with a total batch size of 256 and a learning rate of $10^{-4}$ with Adam.

**DRAGON training settings and hyperparameters.** All DRAGON fine-tuning is performed on top of the baseline model introduced above on four or eight A100 GPUs with a total batch size of 80, with the baseline model used as the "reference model $f_{\mathrm{ref}}$" required by the loss functions, as defined in the $A_\theta$ term in (7). Fine-tuning takes 3-5 days depending on the reward, but we found this can be accelerated by reducing the number of diffusion steps. During DRAGON fine-tuning, the textual conditions are from the pre-training dataset, and no ground-truth audio is used unless required by the reward function. We use Adam with a fixed learning rate of $3 \times 10^{-6}$ and a gradient clip of 45 (determined via gradient logging). For the DPO loss (6) and the KTO loss (8), we select $\beta = 5000$ following (Wallace et al., 2024) and do not update $f_{\mathrm{ref}}$. With paired DPO or KTO, the batch of 80 demonstrations is generated from 40 prompts in pairs, each pair consisting of one positive and one negative demonstration. With unpaired KTO, the 80 demonstrations are from 80 distinct prompts, and the positive/negative label is assigned by comparing each demonstration with the batch mean. All training runs (pre-training and DRAGON) use a 10% condition dropout to enhance classifier-free guidance (CFG). The online audio demonstrations in DRAGON are generated with the default inference setting (40 diffusion steps with the second-order DPM sampler with CFG++ enabled in selected time steps). We use $f_\theta$ to produce the demonstrations with probability 0.9 and use $f_{\mathrm{ref}}$ with probability 0.1. Intuitively, mixing in $f_{\mathrm{ref}}$ generations "anchors" the generation quality to the reference model and provides additional regularization. We use this same configuration for each reward studied, selecting set sizes to fill GPU memory.

# F    Training Loop Pseudocode Walkthrough

We provide an algorithm walkthrough with pseudocode for the most relevant, novel aspects of our DRAGON fine-tuning framework. For simplicity, we consider paired demonstrations. We begin by defining our main training loop that inputs prompts (`prompts`), the model being fine-tuned (`f_theta`), reference model (`f_ref`), the reward function (`reward_function`), and an optional exemplar set (`expemlar_set`).

```
1  def train_step(prompts, f_theta, f_ref, reward_function, exemplar_set):
2
3      # Generate on-policy diffusion latents for two batches of demonstrations
4      with torch.no_grad:
5          embd_1, embd_2 = run_on_policy_inference(prompts)
```

```
6
7      # Decode demonstration latents to audio waveforms to query reward function
8      audio_1, audio_2 = vae_decode(embd_1, embd_2)
9
10     # Use reward function to construct D_+ and D_-
11     if instance_level_reward:
12         D_pos, D_neg = compare_r_instance(
13             audio_1, audio_2, reward_function, exemplar_set
14         )
15     else:
16         D_pos, D_neg = compare_r_dist(
17             audio_1, audio_2, reward_function, exemplar_set
18         )
19
20     # Randomly sample noise levels.
21     # DPO uses the same timesteps for each pair; KTO samples i.i.d.
22     sigmas = noise_distribution_for_embds(embds)
23
24     # Get x_t from x_0 (the demonstrations) by adding noise (forward process)
25     embds_all = torch.stack([embd_1, embd_2])
26     x_noisy = embds_all + sigmas * torch.randn_like(embds)
27
28     # Compute de-noised values f_theta (x_t, t)
29     x_theta = f_theta.denoise(x_noisy, sigmas, prompts)
30
31     # Use the reference model to get f_ref (x_t, t)
32     with torch.no_grad():
33         x_ref = f_ref.denoise(x_noisy, sigmas, prompts)
34
35     # Calculate the KTO/DPO training loss
36     return diffusion_kto_loss(D_pos, D_neg, x_ref, x_theta)
```

Given the main train loop, we further add helper functions for comparing instance-level rewards like $r_{\text{instance}}$ as well as distribution-level rewards like $r_{\text{dist}}$. For reward functions that evaluate individual generations, we input two demonstration sets (`audio_1` and `audio_2`), the reward function, and an optional exemplar set:

```
1  def compare_r_instance(audio_1, audio_2, reward_function, exemplar_set):
2
3      # Compute reward values
4      rewards_1 = reward_function(audio_1, exemplar_set)
5      rewards_2 = reward_function(audio_2, exemplar_set)
6
7      # Element-wise swap for minimization
8      D_pos = torch.where(rewards_1 > rewards_2, audio_1, audio_2)
9      D_neg = torch.where(rewards_1 < rewards_2, audio_1, audio_2)
10
11     return D_pos, D_neg
```

For reward functions that evaluate distributions, we show a piece of psuedocode with GPU parallelization logic. The function `compare_r_dist` handles the parallelization and calls `optimize_r_dist` which implements Algorithm 1.

```
1  def compare_r_dist(
2      audio_1, audio_2, reward_function, exemplar_set, parallelize=True
3  ):
4      batch_size_per_gpu = audio_1.shape[0]
5
6      # Gather all embeddings from all GPUs
7      audio_1, audio_2 = all_gather(audio_1), all_gather(audio_2)
```

```
8        audio_1, audio_2 = audio_1.flatten(0, 1), audio_2.flatten(0, 1)
9        # audio_1 and audio_2 now have shape (# gpus * batch_size_per_gpu, wav_length)
10
11       # Get the indices of the current GPU's embeddings
12       curr_gpu_idx = self.trainer.global_rank
13       idx_curr_gpu = \
14           np.arange(batch_size_per_gpu) + curr_gpu_idx * batch_size_per_gpu
15       # GPU 0 should have 0, ..., batch_size_per_gpu-1
16       # GPU 1 should have batch_size_per_gpu, ..., 2*batch_size_per_gpu-1
17       # etc.
18
19       # If using parallelized optimization, only swap the indices of the current GPU
20       # Otherwise, swap all indices (more accurate but slower)
21       idx_to_swap = idx_curr_gpu if parallelize else np.arange(audio_1.shape[0])
22
23       # Optimize the dataset FAD
24       D_pos, D_neg = optimize_r_dist(
25           audio_1, audio_2, reward_function, exemplar_set, idx_to_swap
26       )
27       return D_pos, D_neg
```

For Algorithm 1 (`optimize_r_dist`), we show an example implementation that optimizes full-dataset FAD as follows. Other rewards like Vendi can be handled similarly.

```
1  def optimize_r_dist(audio_1, audio_2, fad_encoder, exemplar_set, idx_to_swap):
2
3      # Convert everything to an embedding space
4      ref_embds = fad_encoder.encode(exemplar_set)
5      embd_1 = fad_encoder.encode(audio_1)
6      embd_2 = fad_encoder.encode(audio_2)
7
8      # Get mean and covariance of reference embeddings
9      ref_stats = get_mean_and_cov(ref_embds)
10
11     # fad_from_embd computes mean and covariance of the generated embedding set
12     # and computes FAD relative to the reference following Eq.(3)
13     fad_1 = fad_from_embd(embd_1, ref_stats)
14     fad_2 = fad_from_embd(embd_2, ref_stats)
15
16     # Initialize positive and negative sets
17     if fad_score_1 < fad_score_2:
18         # If D_1 is better, we initialize D_+ with D_1 and D_- with D_2
19         D_pos, fad_pos = embds_1, fad_score_1
20         D_neg, fad_neg = embds_2, fad_score_2
21     else:
22         # If D_2 is better, we initialize D_+ with D_2 and D_- with D_1
23         D_pos, fad_pos = embds_2, fad_score_2
24         D_neg, fad_neg = embds_1, fad_score_1
25
26     # Iterative swapping procedure
27     for idx in idx_to_swap:
28         D_pos[idx], D_neg[idx] = D_neg[idx], D_pos[idx]
29
30         # Calculate FAD for the pos/neg sets with the swapped pair
31         new_fad_pos = fad_from_embd(D_pos, ref_stats)
32
33         if new_fad_pos < fad_pos:  # If dataset FAD improved, accept the swap
34             fad_pos = new_fad_pos
35         else:  # If dataset FAD did not improve, revert the swap
```

Table 7: All DRAGON models' instance-level reward win rate.

| DRAGON Model | Aesthetics Score | CLAP Score | Per-Song FAD | | | | | | | |
| | | | CLAP-Audio | | CLAP-Text | | | | VAE-Audio | |
| | | | ALIM | SDNV | ALIM | SDNV | Slackbot | Mixtral | ALIM | SDNV |
|---|---|---|---|---|---|---|---|---|---|---|
| Reference (40 inference steps) | 50.0% | 50.0% | 50.0% | 50.0% | 50.0% | 50.0% | 50.0% | 50.0% | 50.0% | 50.0% |
| Reference (10 inference steps) | 13.9% | 23.8% | 48.4% | 53.0% | 43.9% | 48.3% | 43.0% | 47.8% | 50.5% | 51.7% |
| KTO Aesthetics | 85.2% | 52.2% | 46.4% | 38.9% | 58.9% | 55.1% | 50.2% | 54.3% | 54.9% | 55.3% |
| DPO Aesthetics (40/40 train/inference steps) | 88.9% | 57.1% | 59.8% | 61.9% | 66.8% | 68.2% | 61.8% | 62.6% | 51.2% | 52.8% |
| DPO Aesthetics (40/10 train/inference steps) | 53.5% | 34.2% | 68.1% | 77.2% | 68.4% | 75.7% | 63.6% | 68.4% | 56.6% | 57.9% |
| DPO Aesthetics (10/40 train/inference steps) | 88.2% | 55.1% | 55.8% | 54.1% | 64.1% | 62.4% | 69.2% | 54.8% | 46.2% | 47.4% |
| DPO Aesthetics (10/10 train/inference steps) | 63.9% | 35.7% | 75.0% | 79.6% | 75.7% | 78.2% | 69.9% | 66.5% | 50.5% | 51.8% |
| KTO-Unpaired Aesthetics | 80.0% | 57.1% | 48.2% | 41.1% | 60.8% | 55.0% | 69.4% | 71.9% | 58.7% | 60.1% |
| KTO CLAP Score | 68.7% | 60.1% | 56.3% | 46.5% | 65.2% | 54.2% | 81.8% | 75.7% | 64.9% | 61.4% |
| KTO Per-Song FAD VAE-ALIM-Audio | 78.3% | 50.9% | 84.0% | 76.9% | 83.9% | 81.0% | 75.9% | 70.1% | 93.9% | 94.0% |
| KTO Per-Song FAD VAE-SDNV-Audio | 51.3% | 49.0% | 45.8% | 39.5% | 48.3% | 42.8% | 54.1% | 59.2% | 63.5% | 66.4% |
| KTO Per-Song FAD CLAP-ALIM-Audio | 59.1% | 52.4% | 80.5% | 79.3% | 78.3% | 77.4% | 70.8% | 74.4% | 38.9% | 39.6% |
| KTO Per-Song FAD CLAP-SDNV-Audio | 44.0% | 41.1% | 56.8% | 61.3% | 55.5% | 58.0% | 38.6% | 54.0% | 43.7% | 43.6% |
| KTO Per-Song FAD CLAP-ALIM-Text | 78.3% | 65.4% | 58.8% | 54.2% | 70.4% | 65.9% | 83.5% | 82.8% | 64.0% | 65.7% |
| KTO Per-Song FAD CLAP-SDNV-Text | 49.7% | 55.7% | 61.5% | 54.6% | 63.8% | 58.2% | 56.3% | 70.1% | 57.6% | 57.1% |
| KTO Per-Song FAD CLAP-Slackbot-Text | 52.9% | 60.7% | 59.8% | 60.9% | 64.3% | 63.3% | 76.0% | 79.8% | 38.0% | 41.0% |
| KTO Per-Song FAD CLAP-Mixtral-Text | 65.5% | 52.2% | 46.7% | 48.3% | 60.3% | 60.5% | 67.0% | 74.8% | 52.8% | 52.2% |
| KTO Dataset FAD VAE-ALIM-Audio | 51.4% | 49.7% | 42.2% | 43.0% | 50.4% | 51.5% | 60.0% | 59.3% | 53.5% | 52.7% |
| KTO Dataset FAD VAE-SDNV-Audio | 42.8% | 47.8% | 38.9% | 41.9% | 38.7% | 40.9% | 47.2% | 52.7% | 51.1% | 50.8% |
| KTO Dataset FAD CLAP-ALIM-Audio | 58.3% | 45.7% | 60.6% | 59.2% | 57.3% | 55.3% | 49.1% | 54.5% | 40.9% | 40.1% |
| KTO Dataset FAD CLAP-SDNV-Audio | 47.7% | 48.8% | 65.2% | 71.4% | 61.4% | 65.4% | 49.9% | 57.4% | 30.5% | 31.4% |
| KTO Dataset FAD CLAP-ALIM-Text | 68.8% | 59.5% | 33.0% | 34.2% | 42.7% | 42.4% | 57.2% | 57.4% | 53.5% | 53.2% |
| KTO Dataset FAD CLAP-SDNV-Text | 41.4% | 52.4% | 41.8% | 41.8% | 48.0% | 46.9% | 61.0% | 63.7% | 36.8% | 38.2% |
| KTO Dataset FAD CLAP-Slackbot-Text | 57.4% | 55.8% | 42.0% | 40.3% | 39.9% | 37.5% | 50.9% | 54.7% | 45.2% | 47.5% |
| KTO Dataset FAD CLAP-Mixtral-Text | 64.6% | 61.1% | 37.6% | 33.3% | 50.1% | 43.3% | 74.0% | 69.1% | 51.8% | 54.5% |
| KTO Vendi Score | 21.1% | 14.1% | 6.3% | 5.9% | 2.8% | 2.7% | 5.5% | 9.3% | 6.4% | 6.6% |
| KTO Aesthetics + Vendi Score | 52.1% | 43.4% | 16.1% | 14.7% | 24.9% | 36.8% | 33.0% | 30.5% | 18.5% | 19.8% |

```
36              D_pos[idx], D_neg[idx] = D_neg[idx], D_pos[idx]
37
38      return D_pos, D_neg
```

## G   Example Spectrograms

We show the spectrograms of example model generations in Figures 11, 12, and 13. The DRAGON model that optimizes per-song VAE-FAD with ALIM reference statistics improves the balance over the frequency ranges. The effect of optimizing aesthetics score is less visible from the spectrograms, but our listening tests find reduced artifacts and improved overall music quality.

## H   Full Result Tables

We use five tables to list the reward metrics achieved by all DRAGON models discussed in the paper. First, we present the win rates and the average values of instance-level rewards (aesthetics score, CLAP score, and per-song FAD) in Tables 7 and 8. Next, we present the win rates and the average values of distribution-level rewards like $r_{\text{dist}}$ (full-dataset FAD and Vendi), evaluated in a bootstrapped setting, in Tables 9 and 10. As mentioned in Section 5.1, this means sampling 40-example generation subsets from our 2185-prompt evaluation set with replacement 1000 times, computing the reward metric for each subset, and reporting the average and win rate among these 1000 numbers. Finally, we present the distribution-level rewards evaluated over the full 2185-instance evaluation set without bootstrapping in Table 11. Vendi is computed over per-song average MA embeddings in the full-dataset setting as in the main paper body, but is computed over all MA embeddings without averaging in the bootstrapped setting.

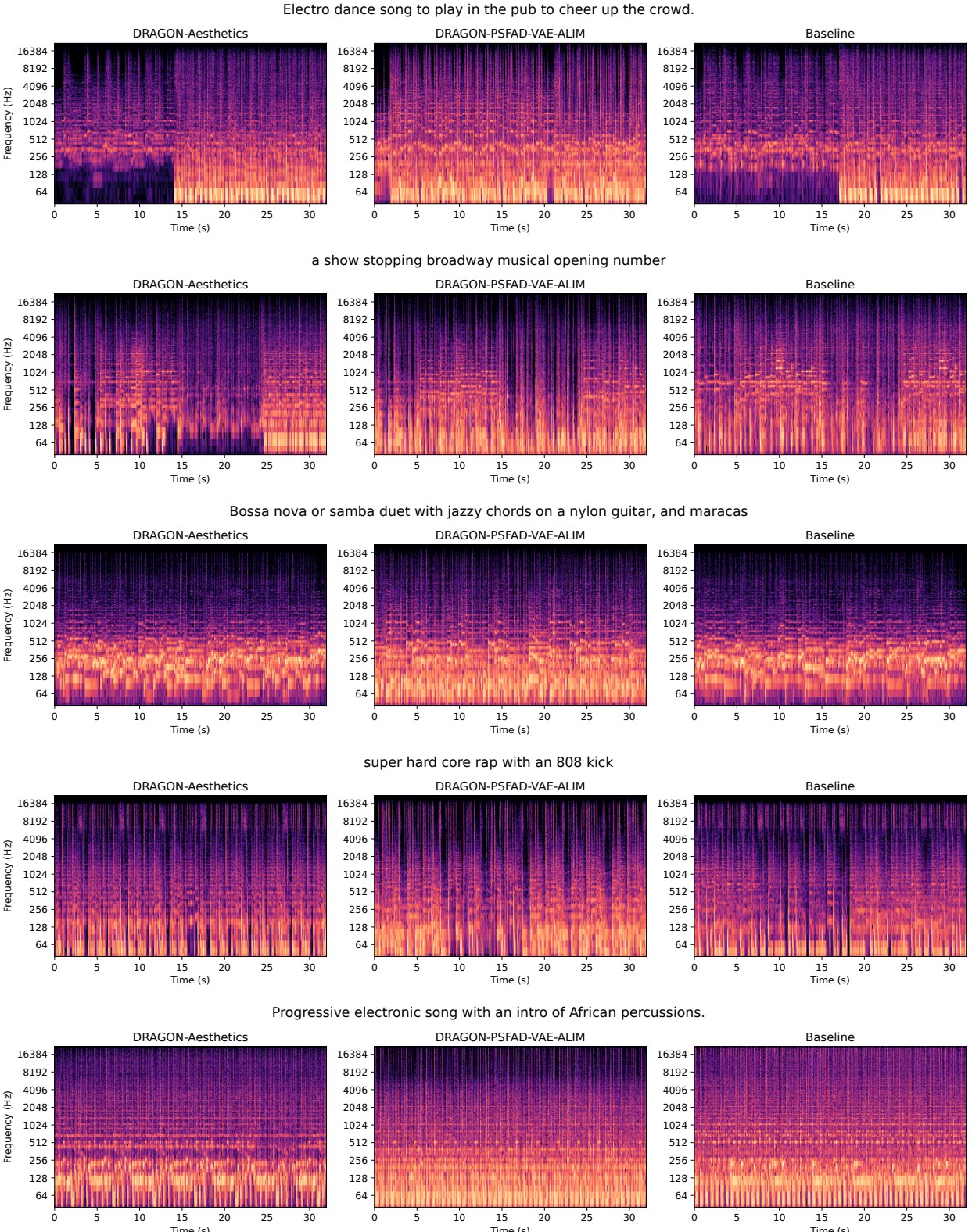

Figure 11: Spectrograms of example generations (part 1).

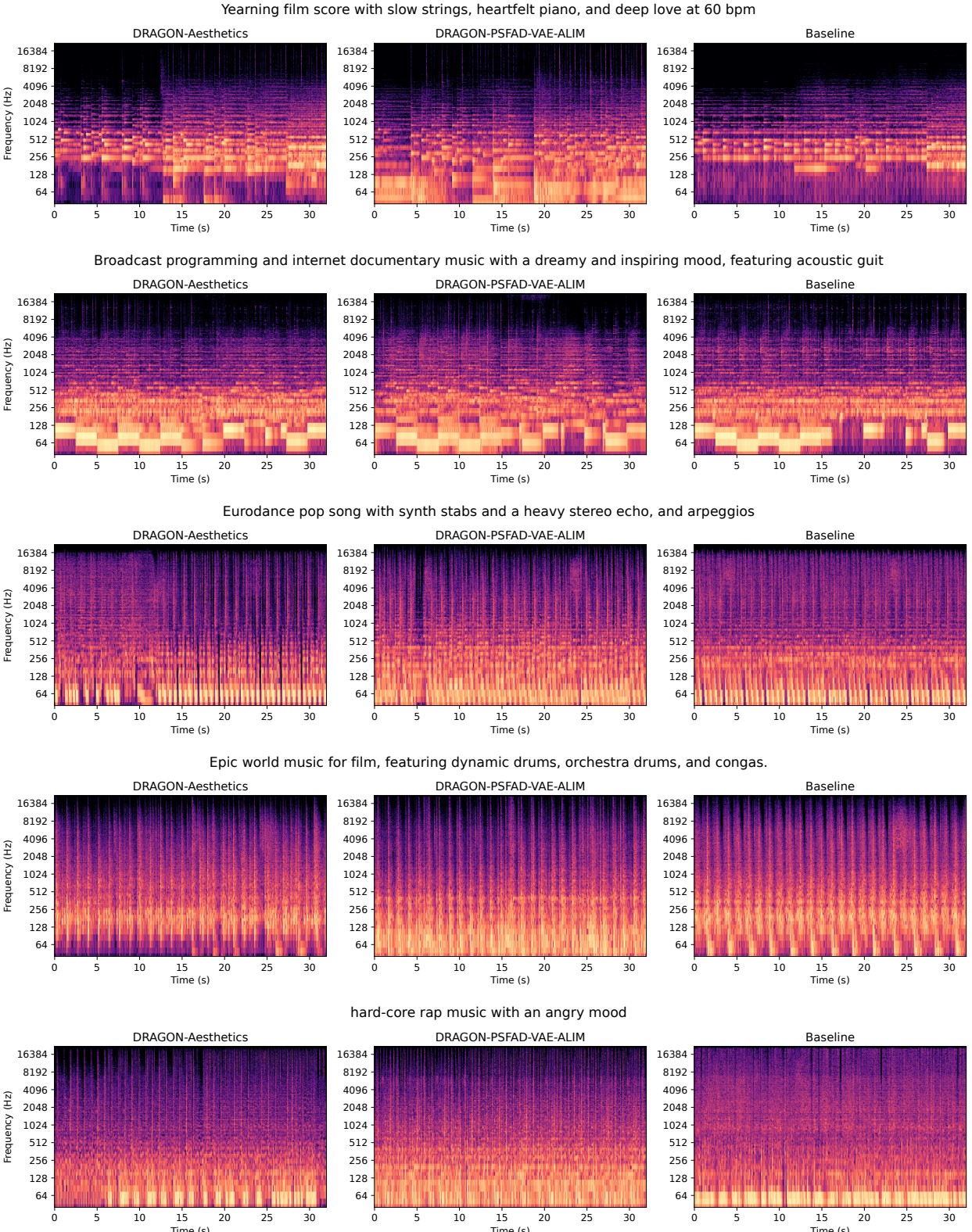

Figure 12: Spectrograms of example generations (part 2).

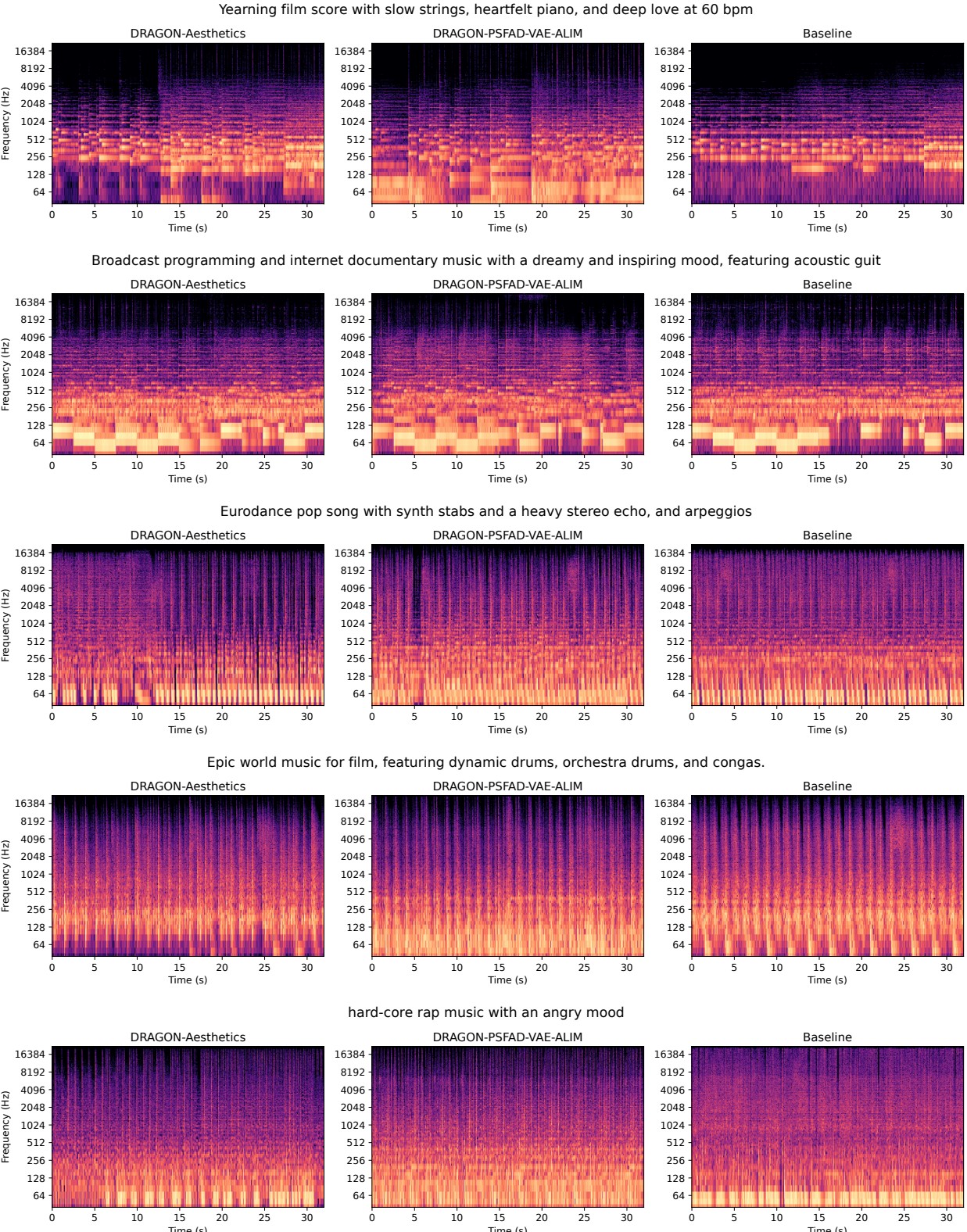

Figure 13: Spectrograms of example generations (part 3).

Table 8: All DRAGON models' average instance-level reward.

| DRAGON Model | Aesthetics Score | CLAP Score | Per-Song FAD | | | | | | | | |
|---|---|---|---|---|---|---|---|---|---|---|
| | | | CLAP-Audio | | CLAP-Text | | | | VAE-Audio | |
| | | | ALIM | SDNV | ALIM | SDNV | Slackbot | Mixtral | ALIM | SDNV |
| Reference (40 inference steps) | 0.187 | 0.300 | 0.947 | 0.990 | 1.576 | 1.561 | 1.596 | 1.534 | 30.84 | 31.12 |
| Reference (10 inference steps) | -0.484 | 0.239 | 0.952 | 0.986 | 1.597 | 1.561 | 1.591 | 1.551 | 30.77 | 30.64 |
| KTO Aesthetics | 0.619 | 0.304 | 0.962 | 1.018 | 1.572 | 1.554 | 1.589 | 1.529 | 29.88 | 30.12 |
| DPO Aesthetics (40/40 train/inference steps) | 0.638 | 0.312 | 0.922 | 0.965 | 1.545 | 1.540 | 1.578 | 1.508 | 30.13 | 30.14 |
| DPO Aesthetics (40/10 train/inference steps) | 0.232 | 0.265 | 0.889 | 0.918 | 1.529 | 1.522 | 1.554 | 1.491 | 28.66 | 28.54 |
| DPO Aesthetics (10/40 train/inference steps) | 0.596 | 0.307 | 0.932 | 0.977 | 1.532 | 1.552 | 1.584 | 1.509 | 30.71 | 30.83 |
| DPO Aesthetics (10/10 train/inference steps) | 0.373 | 0.270 | 0.875 | 0.915 | 1.510 | 1.528 | 1.558 | 1.486 | 30.00 | 29.91 |
| KTO-Unpaired Aesthetics | 0.596 | 0.312 | 0.949 | 1.004 | 1.521 | 1.522 | 1.563 | 1.484 | 29.95 | 30.03 |
| KTO CLAP Score | 0.415 | 0.317 | 0.929 | 0.997 | 1.487 | 1.516 | 1.548 | 1.467 | 27.51 | 28.55 |
| KTO Per-Song FAD VAE-ALIM-Audio | 0.562 | 0.303 | 0.847 | 0.920 | 1.497 | 1.521 | 1.559 | 1.473 | 16.31 | 16.89 |
| KTO Per-Song FAD VAE-SDNV-Audio | 0.102 | 0.278 | 0.990 | 1.040 | 1.621 | 1.572 | 1.610 | 1.568 | 28.99 | 28.38 |
| KTO Per-Song FAD CLAP-ALIM-Audio | 0.291 | 0.304 | 0.867 | 0.923 | 1.522 | 1.515 | 1.550 | 1.480 | 34.55 | 34.52 |
| KTO Per-Song FAD CLAP-SDNV-Audio | 0.120 | 0.286 | 0.935 | 0.973 | 1.598 | 1.555 | 1.600 | 1.542 | 32.98 | 33.31 |
| KTO Per-Song FAD CLAP-ALIM-Text | 0.541 | 0.325 | 0.921 | 0.977 | 1.482 | 1.499 | 1.536 | 1.457 | 27.30 | 27.23 |
| KTO Per-Song FAD CLAP-SDNV-Text | 0.151 | 0.306 | 0.921 | 0.980 | 1.568 | 1.532 | 1.568 | 1.511 | 29.30 | 29.83 |
| KTO Per-Song FAD CLAP-Slackbot-Text | 0.221 | 0.317 | 0.929 | 0.969 | 1.509 | 1.508 | 1.536 | 1.469 | 34.81 | 34.08 |
| KTO Per-Song FAD CLAP-Mixtral-Text | 0.370 | 0.302 | 0.959 | 0.998 | 1.534 | 1.515 | 1.551 | 1.489 | 30.54 | 30.92 |
| KTO Dataset FAD VAE-ALIM-Audio | 0.199 | 0.300 | 0.963 | 1.003 | 1.555 | 1.548 | 1.584 | 1.517 | 31.15 | 31.70 |
| KTO Dataset FAD VAE-SDNV-Audio | 0.107 | 0.296 | 0.976 | 1.010 | 1.590 | 1.560 | 1.600 | 1.539 | 32.17 | 32.46 |
| KTO Dataset FAD CLAP-ALIM-Audio | 0.250 | 0.295 | 0.928 | 0.977 | 1.582 | 1.558 | 1.590 | 1.535 | 32.91 | 33.40 |
| KTO Dataset FAD CLAP-SDNV-Audio | 0.175 | 0.299 | 0.921 | 0.957 | 1.574 | 1.550 | 1.587 | 1.529 | 36.38 | 36.20 |
| KTO Dataset FAD CLAP-ALIM-Text | 0.399 | 0.313 | 0.985 | 1.024 | 1.560 | 1.549 | 1.574 | 1.517 | 30.96 | 31.32 |
| KTO Dataset FAD CLAP-SDNV-Text | 0.091 | 0.304 | 0.965 | 1.005 | 1.552 | 1.543 | 1.577 | 1.515 | 33.37 | 33.24 |
| KTO Dataset FAD CLAP-Slackbot-Text | 0.258 | 0.307 | 0.964 | 1.008 | 1.575 | 1.553 | 1.588 | 1.527 | 32.71 | 32.58 |
| KTO Dataset FAD CLAP-Mixtral-Text | 0.368 | 0.320 | 0.981 | 1.030 | 1.513 | 1.527 | 1.557 | 1.483 | 32.09 | 31.79 |
| KTO Vendi Score | -0.328 | 0.142 | 1.302 | 1.323 | 1.864 | 1.753 | 1.794 | 1.800 | 80.57 | 79.64 |
| KTO Aesthetics + Vendi Score | 0.222 | 0.285 | 1.064 | 1.095 | 1.655 | 1.588 | 1.627 | 1.585 | 41.33 | 41.08 |

Table 9: All DRAGON models' bootstrapped distribution-level reward win rate.

| DRAGON Model | Full-Dataset FAD | | | | | | | | Vendi Diversity Score |
|---|---|---|---|---|---|---|---|---|---|
| | CLAP-Audio | | CLAP-Text | | | | VAE-Audio | | |
| | ALIM | SDNV | ALIM | SDNV | Slackbot | Mixtral | ALIM | SDNV | Score |
| Reference (40 inference steps) | 50.0% | 50.0% | 50.0% | 50.0% | 50.0% | 50.0% | 50.0% | 50.0% | 50.0% |
| Reference (10 inference steps) | 0.0% | 0.0% | 0.0% | 0.0% | 0.0% | 0.0% | 0.0% | 0.0% | 0.0% |
| KTO Aesthetics | 4.3% | 0.0% | 7.1% | 0.9% | 2.7% | 2.8% | 20.1% | 17.4% | 0.0% |
| DPO Aesthetics (40/40 train/inference steps) | 42.4% | 44.8% | 37.3% | 1.1% | 1.6% | 10.3% | 13.3% | 17.1% | 0.0% |
| DPO Aesthetics (40/10 train/inference steps) | 0.0% | 0.0% | 0.6% | 0.0% | 0.0% | 0.0% | 0.2% | 0.2% | 0.0% |
| DPO Aesthetics (10/40 train/inference steps) | 42.1% | 33.8% | 85.0% | 0.1% | 1.0% | 18.4% | 18.4% | 22.3% | 0.0% |
| DPO Aesthetics (10/10 train/inference steps) | 0.3% | 0.8% | 7.7% | 0.0% | 0.0% | 0.1% | 0.0% | 0.1% | 0.0% |
| KTO-Unpaired Aesthetics | 2.6% | 0.3% | 89.5% | 28.5% | 17.3% | 81.5% | 16.5% | 16.8% | 0.0% |
| KTO CLAP Score | 31.7% | 0.2% | 100.0% | 35.1% | 45.3% | 96.3% | 85.2% | 55.3% | 0.0% |
| KTO Per-Song FAD VAE-ALIM-Audio | 38.1% | 0.6% | 45.8% | 0.0% | 0.0% | 1.8% | 86.9% | 85.1% | 0.0% |
| KTO Per-Song FAD VAE-SDNV-Audio | 0.7% | 0.1% | 2.9% | 7.7% | 9.6% | 3.0% | 4.4% | 13.6% | 40.5% |
| KTO Per-Song FAD CLAP-ALIM-Audio | 50.3% | 15.5% | 46.0% | 6.9% | 7.0% | 21.7% | 0.0% | 0.0% | 0.0% |
| KTO Per-Song FAD CLAP-SDNV-Audio | 7.5% | 16.8% | 0.0% | 7.0% | 0.7% | 0.2% | 60.9% | 55.8% | 6.8% |
| KTO Per-Song FAD CLAP-ALIM-Text | 16.0% | 2.0% | 99.8% | 44.5% | 45.5% | 93.4% | 88.5% | 92.0% | 0.0% |
| KTO Per-Song FAD CLAP-SDNV-Text | 43.9% | 8.1% | 16.3% | 67.0% | 53.1% | 42.7% | 29.9% | 17.6% | 0.4% |
| KTO Per-Song FAD CLAP-Slackbot-Text | 1.9% | 6.7% | 98.4% | 74.2% | 89.6% | 95.6% | 0.0% | 0.1% | 0.0% |
| KTO Per-Song FAD CLAP-Mixtral-Text | 0.1% | 0.0% | 55.6% | 40.4% | 40.5% | 50.4% | 83.5% | 79.8% | 0.0% |
| KTO Dataset FAD VAE-ALIM-Audio | 1.0% | 1.5% | 78.9% | 38.3% | 33.9% | 62.8% | 70.5% | 58.7% | 7.3% |
| KTO Dataset FAD VAE-SDNV-Audio | 0.0% | 0.2% | 16.9% | 63.5% | 50.2% | 38.9% | 61.9% | 59.4% | 71.8% |
| KTO Dataset FAD CLAP-ALIM-Audio | 73.6% | 29.9% | 13.8% | 29.8% | 39.4% | 24.0% | 61.5% | 50.2% | 19.8% |
| KTO Dataset FAD CLAP-SDNV-Audio | 42.4% | 83.2% | 13.2% | 15.0% | 15.3% | 11.9% | 0.0% | 0.0% | 6.2% |
| KTO Dataset FAD CLAP-ALIM-Text | 2.1% | 0.9% | 85.4% | 75.2% | 97.8% | 88.2% | 88.2% | 84.7% | 58.3% |
| KTO Dataset FAD CLAP-SDNV-Text | 6.2% | 5.8% | 92.7% | 81.6% | 85.0% | 86.3% | 1.2% | 1.9% | 36.6% |
| KTO Dataset FAD CLAP-Slackbot-Text | 26.2% | 14.1% | 86.0% | 98.8% | 98.4% | 96.5% | 26.5% | 38.9% | 93.2% |
| KTO Dataset FAD CLAP-Mixtral-Text | 1.3% | 0.0% | 100.0% | 89.2% | 95.3% | 99.8% | 8.3% | 14.1% | 7.6% |
| KTO Vendi Score | 0.0% | 0.0% | 0.0% | 0.0% | 0.0% | 0.0% | 0.0% | 0.0% | 99.8% |
| KTO Aesthetics + Vendi Score | 0.0% | 0.0% | 0.0% | 2.9% | 2.8% | 0.1% | 0.0% | 0.0% | 83.1% |

Table 10: All DRAGON models' average bootstrapped distribution-level reward.

| DRAGON Model | Full-Dataset FAD | | | | | | | | Vendi Diversity |
|---|---|---|---|---|---|---|---|---|---|
| | CLAP-Audio | | CLAP-Text | | | | VAE-Audio | | |
| | ALIM | SDNV | ALIM | SDNV | Slackbot | Mixtral | ALIM | SDNV | Score |
| Reference (40 inference steps) | 0.214 | 0.260 | 0.983 | 0.799 | 0.837 | 0.832 | 8.261 | 8.297 | 12.705 |
| Reference (10 inference steps) | 0.344 | 0.374 | 1.069 | 0.876 | 0.904 | 0.925 | 13.360 | 12.502 | 10.401 |
| KTO Aesthetics | 0.243 | 0.311 | 1.004 | 0.831 | 0.863 | 0.859 | 9.135 | 9.182 | 10.968 |
| DPO Aesthetics (40/40 train/inference steps) | 0.216 | 0.262 | 0.987 | 0.826 | 0.864 | 0.848 | 9.616 | 9.288 | 10.601 |
| DPO Aesthetics (40/10 train/inference steps) | 0.295 | 0.321 | 1.029 | 0.876 | 0.906 | 0.899 | 12.604 | 11.844 | 8.736 |
| DPO Aesthetics (10/40 train/inference steps) | 0.217 | 0.265 | 0.969 | 0.832 | 0.865 | 0.843 | 9.245 | 9.022 | 10.731 |
| DPO Aesthetics (10/10 train/inference steps) | 0.262 | 0.297 | 1.006 | 0.876 | 0.904 | 0.888 | 13.657 | 12.798 | 8.710 |
| KTO-Unpaired Aesthetics | 0.242 | 0.303 | 0.963 | 0.806 | 0.850 | 0.819 | 9.703 | 9.560 | 10.664 |
| KTO CLAP Score | 0.222 | 0.309 | 0.926 | 0.804 | 0.838 | 0.806 | 6.955 | 8.135 | 10.516 |
| KTO Per-Song FAD VAE-ALIM-Audio | 0.219 | 0.306 | 0.984 | 0.872 | 0.908 | 0.867 | 7.027 | 7.305 | 8.287 |
| KTO Per-Song FAD VAE-SDNV-Audio | 0.272 | 0.334 | 1.039 | 0.827 | 0.863 | 0.882 | 10.722 | 9.748 | 12.567 |
| KTO Per-Song FAD CLAP-ALIM-Audio | 0.214 | 0.274 | 0.984 | 0.818 | 0.857 | 0.843 | 14.034 | 13.456 | 10.348 |
| KTO Per-Song FAD CLAP-SDNV-Audio | 0.229 | 0.270 | 1.020 | 0.812 | 0.860 | 0.861 | 7.927 | 8.132 | 12.028 |
| KTO Per-Song FAD CLAP-ALIM-Text | 0.227 | 0.290 | 0.937 | 0.801 | 0.838 | 0.812 | 6.879 | 6.822 | 10.048 |
| KTO Per-Song FAD CLAP-SDNV-Text | 0.216 | 0.279 | 0.997 | 0.794 | 0.836 | 0.834 | 8.958 | 9.381 | 11.428 |
| KTO Per-Song FAD CLAP-Slackbot-Text | 0.243 | 0.281 | 0.950 | 0.791 | 0.821 | 0.808 | 12.928 | 11.916 | 11.083 |
| KTO Per-Song FAD CLAP-Mixtral-Text | 0.269 | 0.315 | 0.981 | 0.802 | 0.840 | 0.832 | 7.114 | 7.440 | 10.737 |
| KTO Dataset FAD VAE-ALIM-Audio | 0.239 | 0.283 | 0.973 | 0.802 | 0.841 | 0.828 | 7.579 | 8.137 | 11.944 |
| KTO Dataset FAD VAE-SDNV-Audio | 0.257 | 0.294 | 0.996 | 0.794 | 0.836 | 0.835 | 7.909 | 8.053 | 13.042 |
| KTO Dataset FAD CLAP-ALIM-Audio | 0.207 | 0.265 | 0.995 | 0.803 | 0.839 | 0.839 | 7.923 | 8.273 | 12.325 |
| KTO Dataset FAD CLAP-SDNV-Audio | 0.216 | 0.251 | 0.999 | 0.810 | 0.849 | 0.847 | 14.529 | 13.712 | 11.939 |
| KTO Dataset FAD CLAP-ALIM-Text | 0.243 | 0.292 | 0.967 | 0.791 | 0.813 | 0.816 | 6.980 | 7.353 | 12.824 |
| KTO Dataset FAD CLAP-SDNV-Text | 0.234 | 0.278 | 0.963 | 0.788 | 0.825 | 0.818 | 11.382 | 10.849 | 12.549 |
| KTO Dataset FAD CLAP-Slackbot-Text | 0.223 | 0.272 | 0.968 | 0.775 | 0.813 | 0.811 | 9.014 | 8.639 | 13.436 |
| KTO Dataset FAD CLAP-Mixtral-Text | 0.245 | 0.304 | 0.925 | 0.782 | 0.813 | 0.786 | 10.404 | 9.863 | 11.842 |
| KTO Vendi Score | 0.668 | 0.692 | 1.298 | 1.014 | 1.038 | 1.122 | 41.102 | 39.209 | 16.068 |
| KTO Aesthetics + Vendi Score | 0.340 | 0.370 | 1.066 | 0.831 | 0.868 | 0.886 | 15.192 | 14.801 | 13.292 |

Table 11: All DRAGON models' full-dataset distribution-level reward.

| DRAGON Model | Full-Dataset FAD | | | | | | | | Vendi Diversity |
|---|---|---|---|---|---|---|---|---|---|
| | CLAP-Audio | | CLAP-Text | | | | VAE-Audio | | |
| | ALIM | SDNV | ALIM | SDNV | Slackbot | Mixtral | ALIM | SDNV | Score |
| Reference (40 inference steps) | 0.107 | 0.155 | 0.901 | 0.684 | 0.716 | 0.734 | 7.470 | 7.503 | 12.230 |
| Reference (10 inference steps) | 0.267 | 0.297 | 1.002 | 0.781 | 0.804 | 0.845 | 12.898 | 12.025 | 8.702 |
| KTO Aesthetics | 0.129 | 0.198 | 0.922 | 0.716 | 0.740 | 0.761 | 8.399 | 8.443 | 10.782 |
| DPO Aesthetics (40/40 train/inference steps) | 0.118 | 0.164 | 0.912 | 0.721 | 0.753 | 0.758 | 8.965 | 8.635 | 9.648 |
| DPO Aesthetics (40/10 train/inference steps) | 0.221 | 0.247 | 0.966 | 0.789 | 0.813 | 0.825 | 12.255 | 11.475 | 7.059 |
| DPO Aesthetics (10/40 train/inference steps) | 0.110 | 0.161 | 0.891 | 0.723 | 0.749 | 0.749 | 8.528 | 8.291 | 10.068 |
| DPO Aesthetics (10/10 train/inference steps) | 0.184 | 0.219 | 0.941 | 0.786 | 0.809 | 0.811 | 13.267 | 12.383 | 7.164 |
| KTO-Unpaired Aesthetics | 0.134 | 0.194 | 0.884 | 0.695 | 0.732 | 0.724 | 8.957 | 8.813 | 10.140 |
| KTO CLAP Score | 0.125 | 0.213 | 0.852 | 0.701 | 0.729 | 0.718 | 6.218 | 7.403 | 9.510 |
| KTO Per-Song FAD VAE-ALIM-Audio | 0.135 | 0.223 | 0.919 | 0.781 | 0.811 | 0.789 | 6.764 | 7.030 | 7.061 |
| KTO Per-Song FAD VAE-SDNV-Audio | 0.164 | 0.227 | 0.959 | 0.712 | 0.742 | 0.785 | 10.321 | 9.328 | 12.601 |
| KTO Per-Song FAD CLAP-ALIM-Audio | 0.128 | 0.190 | 0.915 | 0.721 | 0.754 | 0.760 | 13.418 | 12.831 | 8.867 |
| KTO Per-Song FAD CLAP-SDNV-Audio | 0.130 | 0.172 | 0.943 | 0.704 | 0.746 | 0.769 | 7.033 | 7.237 | 11.200 |
| KTO Per-Song FAD CLAP-ALIM-Text | 0.124 | 0.187 | 0.861 | 0.694 | 0.725 | 0.721 | 6.236 | 6.172 | 9.269 |
| KTO Per-Song FAD CLAP-SDNV-Text | 0.115 | 0.179 | 0.920 | 0.686 | 0.722 | 0.742 | 8.295 | 8.720 | 10.718 |
| KTO Per-Song FAD CLAP-Slackbot-Text | 0.145 | 0.184 | 0.875 | 0.685 | 0.709 | 0.718 | 12.202 | 11.187 | 10.085 |
| KTO Per-Song FAD CLAP-Mixtral-Text | 0.166 | 0.213 | 0.903 | 0.693 | 0.725 | 0.739 | 6.408 | 6.727 | 10.083 |
| KTO Dataset FAD VAE-ALIM-Audio | 0.128 | 0.173 | 0.890 | 0.686 | 0.718 | 0.729 | 6.664 | 7.221 | 11.665 |
| KTO Dataset FAD VAE-SDNV-Audio | 0.154 | 0.192 | 0.914 | 0.681 | 0.716 | 0.738 | 6.971 | 7.111 | 12.548 |
| KTO Dataset FAD CLAP-ALIM-Audio | 0.108 | 0.168 | 0.918 | 0.696 | 0.725 | 0.747 | 7.053 | 7.407 | 11.387 |
| KTO Dataset FAD CLAP-SDNV-Audio | 0.119 | 0.154 | 0.922 | 0.703 | 0.735 | 0.756 | 13.905 | 13.065 | 10.943 |
| KTO Dataset FAD CLAP-ALIM-Text | 0.134 | 0.184 | 0.884 | 0.676 | 0.691 | 0.717 | 6.143 | 6.513 | 12.476 |
| KTO Dataset FAD CLAP-SDNV-Text | 0.123 | 0.168 | 0.879 | 0.671 | 0.701 | 0.718 | 10.661 | 10.119 | 12.331 |
| KTO Dataset FAD CLAP-Slackbot-Text | 0.114 | 0.165 | 0.885 | 0.659 | 0.690 | 0.711 | 8.193 | 7.810 | 13.044 |
| KTO Dataset FAD CLAP-Mixtral-Text | 0.130 | 0.191 | 0.842 | 0.664 | 0.688 | 0.686 | 9.564 | 9.014 | 11.669 |
| KTO Vendi Score | 0.546 | 0.571 | 1.200 | 0.879 | 0.895 | 1.007 | 39.049 | 37.143 | 17.225 |
| KTO Aesthetics + Vendi Score | 0.224 | 0.254 | 0.977 | 0.708 | 0.738 | 0.780 | 14.290 | 13.902 | 13.624 |

