# OpenReview forum: "DRAGON: Distributional Rewards Optimize Diffusion Generative Models"
_TMLR — Accepted by TMLR_

### Review · Reviewer_aVwS · 2025-06-02

**Summary Of Contributions:**

This paper propose **DRAGON**,  a flexible framework for fine-tuning diffusion generative models to guide them towards generating desired outcomes.
Compared to traditional reinforcement learning (RL), DRAGON is more flexible and can integrate various types of reward functions, including instance-level, instance-to-distribution, and distribution-to-distribution signals. The authors conducted extensive experiments across multiple tasks and datasets, particularly evaluating 20 different reward functions in the text-to-music generation scenario, and validated the effectiveness of the method through human listening tests.

**Audience:**

Yes

**Claims And Evidence:**

Yes

**Requested Changes:**

Please refer to the weaknesses section for specific suggestions

**Strengths And Weaknesses:**

## Strengths
1. **Novelty**: The paper introduces a new fine-tuning method for diffusion models that is more flexible than existing methods like RLHF or DPO.
2. **Comprehensive Evaluation**: The authors conducted extensive experiments across multiple tasks and datasets, providing a thorough evaluation of the proposed method.
3. **Human-Centric Validation**: The effectiveness of the method was validated through human listening tests, which adds credibility to the results.


## Weaknesses
1. **Unclear Problem Statement**: The introduction does not clearly articulate the shortcomings of existing works and how DRAGON addresses these issues. For example, while the authors mention that standard reward signals in language model training pose challenges in audio/music generation tasks, they do not specify the source of these challenges or explain how DRAGON overcomes them.
2. **Limited Generalization Evidence Beyond Music**: Although the paper claims that the design is modality-agnostic, there is a lack of experiments on image or video generation tasks. For instance, applying DRAGON to image generation tasks or using video tasks would enhance its generalization capabilities.
3. **Lack of Comparison with Existing Methods**: The experimental section does not compare with existing reinforcement learning methods, such as DPO or RLHF. Although the authors claim that DRAGON outperforms these methods in terms of flexibility and adaptability, the lack of direct comparative experiments makes it difficult to validate this point.

---

> ### Author Response · Authors · 2025-08-26
> **Thank you for your feedback**
>
> Thank you for your comments and suggestions. They are extremely helpful for us to clarify our results and findings. Following the suggestions, we have updated our paper, with the modifications highlighted in blue.
>
> ### Q1. Problem Statement.
>
> **Q1.** Unclear Problem Statement: The introduction does not clearly articulate the shortcomings of existing works and how DRAGON addresses these issues. For example, while the authors mention that standard reward signals in language model training pose challenges in audio/music generation tasks, they do not specify the source of these challenges or explain how DRAGON overcomes them.
>
> **A1.** Thank you for letting us know the potential ambiguities in our descriptions. We present a more detailed explanation of the challenges DRAGON aims to solve.
> - **Scarcity of human preference annotations.** Media like audio, music, and video are multi-modal and highly perceptual, making it hard and expensive to create reliable preference pairs. This is because music preferences are largely subjective, with different listeners resonating with different genres/melodies. Furthermore, labeling long-form audio/music can be more cumbersome than textual and visual signals -- assigning a preference score for a 32-second audio demands 32-second listening time, whereas text/images can be more easily skimmed through.
> - **Challenges in criteria-based rewards.** Research in language models considered implementing criteria-based reward signals to alleviate the preference data challenges [1]. However, similar approaches have been scarce for media creation, because its evaluation metrics often measure distributional properties like Frechet embedding distance, diversity, and coverage, which existing reward optimization approaches like reinforcement learning cannot handle.
> - **Rewards beyond human preference.** Furthermore, preference-based training methods like DPO directly learn from large-scale paired datasets, with the reward function (human preference) implicitly encoded. They are thus constrained by this implicit reward hidden in the training data. For reward functions other than human preference, DPO-like methods demand collecting new large-scale datasets to represent those rewards, complicating their applications. For modalities like music, where human preference datasets are scarce, alternative reward functions are not hypothetical scenarios -- they are necessities.
>
> DRAGON tackles these challenges via two key innovations.
> - **Distributional reward optimization.** Unlike existing methods that can only deal with reward functions that assign a value to each generation, DRAGON can optimize reward functions that evaluate distributions via the demonstrative distributions $\mathcal{D}\_+$ and $\mathcal{D}\_-$.
> - **Distribution-matching reward functions.** We propose to use reward functions like CLAP score as well as per-song and full-distribution Frechet audio distance to match distributions on instance-to-instance, instance-to-distribution, and distribution-to-distribution levels. By optimizing these rewards with DRAGON, we can improve human-perceived generation quality without human preference annotations.
> These two innovations enhance each other, effectively addressing the above challenges.
>
> **In summary, DRAGON tackles challenges in optimizing distributional reward functions which are powerful for media generation, thereby overcoming the lack of human preference data in such modalities.**

---

> > ### Comment · Reviewer_aVwS · 2025-08-30
> > **Thanks for the authors' response**
> >
> > Thanks for the author's response, which highlights the challenges of the studied problem clearly and provides more explanations and comparisons with prior works. My main concerns are addressed. Overall, the paper proposes a new reward-based training method for diffusion models and contains extensive experiments. Therefore, I recommend the acceptance of this paper.

---

> > > ### Author Response · Authors · 2025-09-03
> > > **Thank you for your response.**
> > >
> > > Thank you for the encouraging response and for recommending acceptance. We are glad that our explanations helped clarify the contribution of our work.

---

> ### Author Response · Authors · 2025-08-26
> **Thank you for your feedback part 2**
>
> ### Q2. Music-Domain Experiments.
>
> **Q2.** Limited Generalization Evidence Beyond Music: Although the paper claims that the design is modality-agnostic, there is a lack of experiments on image or video generation tasks. For instance, applying DRAGON to image generation tasks or using video tasks would enhance its generalization capabilities.
>
> **A2.** Thank you for the suggestions.
> Following the precedent set by foundational works in this area, such as RLHF [2], DPO [3] (language), Diffusion-DPO [4], and DDPO [5] (image), we chose to focus on a single, challenging modality. This experiment design allowed us to conduct a deep and comprehensive evaluation of DRAGON's capabilities across 20 distinct reward functions, where we observe consistent improvements. We specifically selected music for our experiments because it is a representative perceptive modality where human preferences are scarce (no suitable open-source preference datasets). Such a scenario demands aligning generations to human preferences with sparse or no human annotation, a challenge that DRAGON is able to solve.
>
> While our experiments focus on enhancing music generation, our proposed methods do not theoretically rely on any properties unique to music. This is because DRAGON operates in a diffusion latent space. Whether the latent space is bridged with generations via a music/audio VAE or an image VAE does not affect the internal workings of DRAGON.

---

> ### Author Response · Authors · 2025-08-26
> **Thank you for your feedback part 3**
>
> ### Q3. Comparison with Existing Methods.
>
> **Q3.** Lack of Comparison with Existing Methods: The experimental section does not compare with existing reinforcement learning methods, such as DPO or RLHF. Although the authors claim that DRAGON outperforms these methods in terms of flexibility and adaptability, the lack of direct comparative experiments makes it difficult to validate this point.
>
> **A3.** We believe there may be a misunderstanding regarding the relationship between DRAGON and existing literature. We apologize for the potential ambiguities in the paper's descriptions, and then clarify DRAGON's relationship with DPO and RLHF below.
>
> **Relationships with DPO:** DPO [3] proposed to directly optimize generative models to human preference, without an explicit reward function. There are several implications. First, while› effective in aligning generations with human preferences, DPO cannot optimize general reward functions. In contrast, DRAGON can optimize arbitrary reward functions, including those that score each generation and those that assign a single value to a distribution of generations. Second, DPO requires large-scale human preference datasets, which are prohibitively expensive to collect, especially for perceptual modalities like music. In contrast, DRAGON improves human-perceived generation quality via sparse human annotations or no preference data at all via distributional rewards. That is, **DRAGON excels in situations where DPO does not apply.** Finally, **DRAGON can use DPO as a loss function (see Section 3.2), and thus should not be viewed as a direct competition.**
>
> **Relationships with RL:** Because DRAGON is a reward optimization framework, it resembles reinforcement learning (RL) from this perspective. While DRAGON does not assume dynamics, many RL with human feedback (RLHF) methods also do not. **DRAGON's difference from existing RLHF settings lies in how they handle diffusion processes and compatibilities with distribution-level reward functions.** Examples of prior RL methods designed for diffusion models include DDPO [5] and DPOK [6]. They generally represent diffusion processes as Markov decision processes (MDPs). These methods are generally evaluated by training on a small set of (<150) prompts constructed via a template, whereas DRAGON achieves improvements over a wide range of training/evaluation prompts over several datasets. Moreover, DRAGON extends beyond human preference to distributional rewards (traditional RL generally mandates instance-wise reward), which are universal in generative models. In other words, **DRAGON can solve problems that traditional RLHF cannot tackle**.
>
> **Therefore, DRAGON does not directly compete with DPO or RLHF. It incorporates DPO as a loss function, and can solve problems that neither DPO nor RLHF can deal with. Hence, we believe that DRAGON is indeed more flexible/versatile than existing works.** In the following table, which is also added to the revised paper, we summarize this comparison. Once again, we apologize for the ambiguities in the original manuscript. In the revised manuscript, we have added clarifications regarding this matter.
>
> | Category               | Example Methods for Diffusion     | Explicit Reward | Distributional Reward |
> |------------------------|-----------------------------------|-----------------|-----------------------|
> | RLHF                   | DDPO, DPOK                        | Yes             | No                    |
> | Preference Optimization| Diffusion-DPO, Diffusion-KTO      | No              | No                    |
> | DRAGON (ours)          | DRAGON                            | Yes             | Yes                   |
>
> [1] DeepSeek-AI. DeepSeek-R1: Incentivizing reasoning capability in LLMs via reinforcement learning. \
> [2] Ziegler et al. Fine-Tuning Language Models from Human Preferences. \
> [3] Rafailov et al. Direct Preference Optimization: Your Language Model is Secretly a Reward Model. \
> [4] Wallace et al. Diffusion Model Alignment Using Direct Preference Optimization. \
> [5] Black et al. Training Diffusion Models with Reinforcement Learning.\
> [6] Fan et al. DPOK: Reinforcement Learning for Fine-tuning Text-to-Image Diffusion Models.

---

### Review · Reviewer_xyNG · 2025-06-30

**Summary Of Contributions:**

The paper proposes a new framework for reward-based optimization of diffusion models, called DRAGON. The approach leverages reference examples and reward functions to come up with positive and negative sets for model finetuning. The approach is compatible with diverse types of rewards, grounded and ungrounded instance-wise as well as distributional metrics. The evaluation, that includes human evaluations, shows that the approach is leading to improvements over vanilla model sampling.

**Audience:**

Yes

**Broader Impact Concerns:**

No concerns.

**Claims And Evidence:**

Yes

**Requested Changes:**

The paper is in quite decent shape already, my main concern is the positioning of the contribution. Overall, I find that Tthe paper would benefit from improved positioning w.r.t. prior art as mentioned in Weaknesses.

**Strengths And Weaknesses:**

Strengths:
- The paper deals with an important problem of model finetuning and is of interest for TMLR community.
- Human listening study adds to the evaluation strength of the paper.
- Showing the results on multiple rewards show that the method applies to numerous finetuning setups/signals.

Weaknesses:
- The presentation of the paper could be improved, e.g., it is a bit unclear what is the main differentiator of Dragon. The reviewer read the introduction several time and it is still unclear what is the differentiator of Dragon w.r.t. other approaches, “DRAGON offers an alternative to existing reinforcement learning (RL) or pair-wise preference approaches…” alternative in which way? Is the use of positive/negative demonstrative distributions claimed to be novel?  In general, the intro would benefit from some re-writing focusing on Dragon’s novelty.
- In the evals, Table 1, there are no direct comparison to RLHF/GRPO. How would Dragon win rate look like if we were to add more finetuning models?
- The evaluation would be stronger if additional modality would be added, e.g. images.

Minor:
The figure on page 2 is missing caption, it has only 2 subcaptions.

---

> ### Author Response · Authors · 2025-08-26
> **Thank you for your feedback**
>
> Thank you for your comments and suggestions. They are valuable for helping us improve our positioning and clarity.
>
> ### Q1. Novelty and Relation To Prior Work
>
> **Q1.** The presentation of the paper could be improved, e.g., it is a bit unclear what is the main differentiator of Dragon. The reviewer read the introduction several times and it is still unclear what is the differentiator of Dragon w.r.t. other approaches, “DRAGON offers an alternative to existing reinforcement learning (RL) or pair-wise preference approaches…” alternative in which way? Is the use of positive/negative demonstrative distributions claimed to be novel?
>
> **A1.** Thank you for this feedback and opportunity to clarify our presentation. The key differentiator of DRAGON is its flexibility to optimize generative models using a broader class of reward functions than existing methods. This is particularly important for perceptual media like music, where human preference data is scarce and quality is often measured by holistic, distributional metrics.
>
> DRAGON's novelty is a product of two key ideas:
>
> - **Optimization of Distributional Rewards.** Unlike methods like RLHF [2], which are limited to rewards which assign a single score to a single output (instance level), **DRAGON can directly optimize for distributional rewards**. These include non-decomposable metrics like Frechet Distance [6], Vendi diversity [7], and other coverage metrics, which evaluate the quality of an entire set of generations. Such metrics are critical in the evaluation of generative model but are incompatible with standard fine-tuning approaches. We emphasize that directly **optimizing per-song FAD and dataset FAD, which we found to be especially suitable for scenarios where human feedback is scarce and AI-automated evaluation is unavailable, is a novel contribution**.
> - **Demonstrative Distributions.** The use of positive and negative demonstrative distributions is the core mechanism that enables this flexibility. By constructing these sets based on a given reward function, DRAGON translates the complex goal of optimizing a distributional metric into a more tractable distribution-matching problem.
>
> In short, DRAGON provides a new path for aligning generative models in domains like music, where collecting preference pairs is difficult, and evaluation depends on metrics that RLHF and DPO cannot handle. We have updated the introduction to highlight these points (shown in blue) and have made clarifications on how demonstrative distributions enable optimizing distributional rewards in Section 3 (also in blue).
>
>
> ### Q2. Comparison With Existing Methods
> **Q1.** In the evals, Table 1, there are no direct comparison to RLHF/GRPO. How would Dragon win rate look like if we were to add more fine-tuning models?
>
> **A1.** We appreciate this question as it highlights a point we need to clarity. DRAGON is not a direct competitor to RLHF/GRPO. DRAGON is a framework that complements and extends RLHF/GRPO-style. The distinction lies in the types of reward signal these approaches can optimize.
>
> - **Preference Optimization.**  Methods like **RLHF/GRPO [1,2] are designed for instance-wise rewards**, such as a human preference score for a single generative example. They cannot optimize for rewards that assess a whole distribution of samples.
> - **Demonstrative Distributions.** DRAGON is **designed to handle instance-wise rewards and distribution-level rewards**. This generality is essential for many generative modeling tasks where the best measures of quality (like Frechet Distance) are distributional.
>
> Because DRAGON can solve a broader class of problems, a direct "win-rate" comparison would be misleading. Its purpose is to tackle reward functions that are out of scope for standard RLHF and DPO. To make this relationship more explicit, we have added the following comparison table to our paper in section 2.3 and more explicitly positioned our task in the introduction.
>
> | Category               | Example Methods for Diffusion     | Explicit Reward | Distributional Reward |
> |------------------------|-----------------------------------|-----------------|-----------------------|
> | RLHF                   | DDPO, DPOK                        | Yes             | No                    |
> | Preference Optimization| Diffusion-DPO, Diffusion-KTO      | No              | No                    |
> | DRAGON (ours)          | DRAGON                            | Yes             | Yes                   |

---

> ### Author Response · Authors · 2025-08-26
> **Thank you for your feedback part 2**
>
> ### Q3. Music-Domain Experiments.
> **Q1.** The evaluation would be stronger if additional modality would be added, e.g. images.
>
> **A1.** We agree that multi-modality would be valuable. Our decision to focus on a single challenging modality was deliberate and allowed us to perform a thorough evaluation.
>
> Following the precedent set by foundational works in alignment, which also focus on a single modality (RLHF/DPO on language [2,3], diffusion approaches on images [4,5]), this approach enabled us to test our framework across more than 20 different rewards. This evaluation shows consistent performance gains and verifies DRAGON's efficacy.
>
> We specifically chose music because it is a representative percetual modality that illustrates the kinds of challenges DRAGON is designed to solve. First, human preference datasets are scarce and challenging to create. Second, distributional metrics are fundamental and essential for evaluating generative model quality. While our experiments focus on music, DRAGON is modality-agnostic by design because it operates in the diffusion model's latent space. The core mechanisms are independent of whether the latent diffusion model decodes into audio or image spaces. We hope to explore other modalities in future work.
>
> [1] DeepSeek-AI. DeepSeek-R1: Incentivizing reasoning capability in LLMs via reinforcement learning. \
> [2] Ziegler et al. Fine-Tuning Language Models from Human Preferences. \
> [3] Rafailov et al. Direct Preference Optimization: Your Language Model is Secretly a Reward Model. \
> [4] Wallace et al. Diffusion Model Alignment Using Direct Preference Optimization. \
> [5] Black et al. Training Diffusion Models with Reinforcement Learning. \
> [6] Kilgour et al. Fréchet Audio Distance: A Metric for Evaluating Music Enhancement Algorithms \
> [7] Friedman et al. The Vendi Score: A Diversity Evaluation Metric for Machine Learning

---

### Review · Reviewer_59me · 2025-08-15

**Summary Of Contributions:**

The paper proposes a distributional reward optimization framework, DRAGON, for online reinforcement learning with a focus on audio-generation diffusion models. DRAGON achieves this by gathering online generations, scoring them to construct two distinct sets, and maximizing the reward contrast between the two sets. DRAGON achieves significant performance improvement over existing alignment techniques without explicitly collecting human-annotated data or preferences.

**Audience:**

Yes

**Claims And Evidence:**

Yes

**Requested Changes:**

Please respond to the weakness section above.

**Strengths And Weaknesses:**

*Strengths*:

1. The paper proposes a novel approach to annotate data on the fly for online reinforcement learning of diffusion models.
2. The proposed framework, DRAGON, works with both instance- and distribution-level rewards.
3. The paper demonstrates the effectiveness of the proposed approach on a variety of benchmark datasets and human evaluations.

*Weaknesses*:

1. The notations in the paper could be improved for clarity.
    * Equation 1 needs more clarification. $r_{dist}$ was defined as a function that takes as input a set of generations. However, Eq. 1 provides a distribution as input. Did the paper mean to input a sample set from that distribution?
    * The following lines are unclear. How do we recover the instance-level reward function? $r_{instance}$ wasn’t even defined before this.
    * Eq. 2 is simply cross-entropy loss, right? The paper can present it in a simplified manner. I'm also confused if this loss was used as the proposed approach or if DPO/KTO was used?

2. The core contribution in Algorithm 1 is confusing to me.
    * I think the variable $i$ is used both for iteration count and selecting the $i$-th sample from the datasets.
    *  Under the current formulation, why can't we construct the subsets D+ and D_ by simply iterating over the datasets and selecting the response with the higher reward? How would this be different from using instance-level rewards?

---

> ### Author Response · Authors · 2025-08-26
> **Thank you for your feedback.**
>
> We thank the reviewer for their detailed feedback. The comments helped us identify areas where clarity and exposition can be improved. We address each point below.
>
> ## Q1 Notation and clarity
>
> ### P1. Equation 1
>
> Equation 1 needs more clarification. $r_{\textrm{dist}}$ was defined as a function that takes as input a set of generations. However, Eq. 1 provides a distribution as input. Did the paper mean to input a sample set from that distribution?
>
> ### A1.
>
> Thank you for highlighting this. You are correct that the reward function is computed on finite sets of samples drawn from the model's output distribution \$\mathcal{D}\_\theta\$.
>
> While we use $\mathcal{D}\_\theta$ as a formal distribution for notational brevity (metrics like FAD and Vendi theoretically operate on distributions), **in practice we operate on sampled batches**.  We have added an explicit clarification in Section 3.1 immediately after Eq. 1: "While the notations $\mathcal{D}\_1$, $\mathcal{D}\_2$, $\mathcal{D}\_+$, and $\mathcal{D}\_-$ technically represent distributions of generations, practical computations of $r_{\textrm{dist}}$ use sampled sets as approximations. Hence, with a slight abuse of notation, we also use these notations to denote the sampled demonstration sets."
>
>
> ### P2. Instance-level reward function
> The following lines are unclear. How do we recover the instance-level reward function? $r_{\textrm{instance}}$ wasn’t even defined before this.
>
> ### A2.
>
> We apologize for the confusion caused by notational ambiguities. The high-level distributional reward optimization objective in Equation 1 includes traditional instance-level optimization widely studied in RLHF as a special case. Specifically, optimizing the distributional reward $r\_\textrm{dist}$ is equivalent to instance-level optimization when $r\_\textrm{dist}$ is the expectation of some instance-level reward $r\_\textrm{instance}$, that is, $r\_\textrm{dist} (\mathcal{D}\_\theta) = \mathbb{E}\_{X \sim \mathcal{D}\_\theta} [r\_\textrm{instance} (X)]$. Here, the notation $r\_\textrm{instance}$ is introduced conceptually to illustrate this special case. It represents an arbitrary instance-level reward that evaluates individual generations, with examples including human or AI-provided preference scores assigned to each generation. In practice, by construction of this special case, $r\_\textrm{instance}$ is readily available and there is **no need to "recover" it** from $r\_\textrm{dist}$. We can then build the positive and negative demonstration sets via simple procedures such as comparing the $r\_\textrm{instance}$ of each generation with some threshold.
> In contrast, beyond this decomposable special case, i.e., $r\_\textrm{dist}$ is not an expectation, general distribution-level rewards cannot be decomposed into instance-level reward functions.
> We have revised the description right after Eq. 1 (highlighted in teal), clarified the $r\_\textrm{instance}$ notation, and removed the potentially misleading word "recover", in order to better explain the instance-level special case.
>
>
> ### P3. Cross-entropy loss
> Eq. 2 is simply cross-entropy loss, right? The paper can present it in a simplified manner. I'm also confused if this loss was used as the proposed approach or if DPO/KTO was used?
>
> ### A3.
> We agree with the reviewer that maximizing the likelihood in Eq. 2 is equivalent to minimizing a cross-entropy loss. We used the likelihood form to align with our implementation perspective. Critically, for diffusion models, the likelihood is implicit and the KL divergence cannot be directly computed. Therefore, **we use surrogate loss functions to indirectly optimize the conceptual cross-entropy** loss, increasing the likelihood of preferred samples and decreasing that of dispreferred ones. Indeed, Diffusion-DPO/KTO loss functions were derived as variants of the cross-entropy loss that handle this likelihood implicitness in the context of fine-tuning generative diffusion models. We have made this explicit in Section 3.2 by adding (highlighted in blue): "Diffusion models pose a greater optimization challenge than auto-regressive ones because $\pi_\theta (\cdot)$ is implicit, making directly optimizing the KL-based conceptual objective impractical. To this end, DRAGON steers the likelihoods via surrogate loss functions, including but not limited to the Diffusion-DPO loss and the Diffusion-KTO loss."

---

> ### Author Response · Authors · 2025-08-26
> **Thank you for your feedback part 2**
>
> ## Q2. Exposition of Alg. 1
>
> ### P1. Iteration count
> I think the variable $i$ is used both for iteration count and selecting the $i$-th sample from the datasets.
>
> ### A1.
> Thank you for pointing this out. Your understanding is correct: the variable $i$ serves as **both the iteration counter and the sample index**. At iteration $i$, we consider whether to swap sample $i$ in our positive/negative sets, thereby evaluating a potential swap for every item in the batch. To improve clarity, we have updated Section 3.1 (highlighted in blue): "Because $\mathcal{D}\_1$ and $\mathcal{D}\_2$ are random roll-outs during training, we do not further randomize and swap the $i^{\text{th}}$ pair at the $i^{\text{th}}$ iteration."
>
> ### P2. Construction of $\mathcal{D}\_+$ and $\mathcal{D}\_-$
> Under the current formulation, why can't we construct the subsets $\mathcal{D}\_+$ and $\mathcal{D}\_-$ by simply iterating over the datasets and selecting the response with the higher reward? How would this be different from using instance-level rewards?
>
> ### A2.
> This is a great question and gets at why Alg. 1 is necessary. The "pick the higher score" approach works for decomposable, instance-level rewards like aesthetics score or CLAP similarity score, where each item is scored independently. Indeed, **our experiments with decomposable rewards use this simple instance-wise comparison**. However, this approach fails for true distributional rewards like full-dataset FAD or Vendi, that evaluate a distribution/set as a whole. In these non-decomposable cases, a single score is assigned to an entire distribution/set, and an explicit reward value for each example is unavailable. That is, the contribution of a single sample depends on the rest of the set. For example, adding an individually "good" sample may change embedding diversity and thereby hurt FAD. **Algorithm 1 is a greedy procedure designed for such non-decomposable rewards**, enabling the iterative construction of $\mathcal{D}\_+$ by swaps that are guaranteed to improve a set-level objective. We have clarified in Section 3.1 (highlighted in teal) that Alg. 1 is introduced specifically to handle non-decomposable metrics, and that a simple instance-wise comparison can be performed for composable rewards.

---

### Author Response · Authors · 2025-08-26
**We have updated our PDF and responded to reviewers**

Dear Esteemed Reviewers and Action Editor,

We would like to thank you for the thoughtful feedback and comments. We have replaced our PDF submission with a new draft (includes highlighted text changes). In our latest submission, we have addressed the reviewers' comments in detail and are responding individually. Overall, we have clarified our contributions, clarified our main algorithm equation details, provided more context of our work compared to RLHF and preference optimization methods such as Diffusion-DPO, and finally justified our focus on in-depth experiments for a single modality as is common practice in the topic area.

Sincerely,
Authors

---

### Decision · Action_Editor_FGnW · 2025-10-09

**Recommendation:** Accept with minor revision

**Additional Comments:**

I recommend minor revisions focused on clarity, positioning, and reproducibility—not new major experiments.

Requested changes:
- Notation & equations.
    - Make explicit throughout that distributional rewards are computed on finite sampled sets; keep the “slight abuse of notation” caveat near Eq.1 and refer back when symbols reappear.
    - Define the instance-level reward r_inst before invoking the expectation special case; remove any “recover” phrasing to avoid implying inversion from distributional rewards.
    - Present Eq. 2 as cross-entropy (or likelihood) cleanly and state clearly that Diffusion-DPO/KTO are surrogate objectives used because likelihood is implicit in diffusion models.
- Algorithm 1 clarity.
    - Disambiguate the variable used for iteration vs. indexing (noted by Reviewer 59me), and add one sentence explaining why greedy swaps are required for non-decomposable rewards and how this differs from the trivial instance-wise selection used for decomposable rewards.
- Presentation polish.
    - Ensure all figures have full captions; fix the page-2 figure issue noted by Reviewer xyNG; tighten the opening paragraph to clearly state problem, gap, and contribution.
- Reproducibility.
    - Provide complete training details (datasets/splits, prompts, batch sizes, compute budget, seeds, hyperparameters for each surrogate loss), and specify how set sizes are chosen for distributional reward estimation.
    - If possible, release code or an artifacts checklist enabling replication of at least one DRAGON configuration with a distribution-level reward (e.g., FAD/Vendi).

**Audience:**

Yes

**Audience Explanation:**

TMLR readers working on generative modeling, diffusion model alignment, and reward/feedback learning will find the contribution relevant. Optimizing non-decomposable, distribution-level objectives is timely and of practical interest beyond music (e.g., image/video diversity/coverage metrics). Even with single-modality experiments, the framework and analysis are broadly applicable.

**Claims And Evidence:**

Yes

**Claims Explanation:**

The paper proposes DRAGON, a framework for fine-tuning diffusion models with both instance-level and genuinely distribution-level rewards. The empirical evidence is substantial within the stated scope (text-to-music): evaluations span >20 reward functions, include human listening studies, and consistently show improvements over baselines aligned to each reward type. Reviewers aVwS and xyNG both judged the main claims to be supported; remaining concerns are largely about exposition/positioning rather than the core technical validity. The authors’ rebuttal clarifies Eq. 1 (samples vs. distributions), the special case reducing to instance-level rewards, and the role of surrogate losses (e.g., Diffusion-DPO/KTO) when likelihood is implicit. While validation is limited to a single modality and direct head-to-head against RLHF/GRPO is not central (given DRAGON’s different objective), the evidence is sufficient for TMLR provided the authors address clarity issues noted by Reviewer 59me.